# Multisensory correlation computations in the human brain identified by a time-resolved encoding model

Jacques Pesnot Lerousseau [1,2,3✉], Cesare V. Parise[4], Marc O. Ernst[2] & Virginie van Wassenhove [3]

Neural mechanisms that arbitrate between integrating and segregating multisensory information are essential for complex scene analysis and for the resolution of the multisensory correspondence problem. However, these mechanisms and their dynamics remain largely unknown, partly because classical models of multisensory integration are static. Here, we used the Multisensory Correlation Detector, a model that provides a good explanatory power for human behavior while incorporating dynamic computations. Participants judged whether sequences of auditory and visual signals originated from the same source (causal inference) or whether one modality was leading the other (temporal order), while being recorded with magnetoencephalography. First, we confirm that the Multisensory Correlation Detector explains causal inference and temporal order behavioral judgments well. Second, we found strong fits of brain activity to the two outputs of the Multisensory Correlation Detector in temporo-parietal cortices. Finally, we report an asymmetry in the goodness of the fits, which were more reliable during the causal inference task than during the temporal order judgment task. Overall, our results suggest the existence of multisensory correlation detectors in the human brain, which explain why and how causal inference is strongly driven by the temporal correlation of multisensory signals.

[1] Aix Marseille Univ, Inserm, INS, Inst Neurosci Syst, Marseille, France. [2] Applied Cognitive Psychology, Ulm University, Ulm, Germany. [3] Cognitive Neuroimaging Unit, CEA DRF/Joliot, INSERM, CNRS, Université Paris-Saclay, NeuroSpin, 91191 Gif/Yvette, France. [4] Independent researcher, Ulm, Germany. ✉email: jacques.pesnot-lerousseau@univ-amu.fr

The brain can integrate information coming from different sensory modalities in a unified percept: seeing a face pronouncing the syllable "ka" dubbed with the sound "pa" yields a distinct multisensory percept "ta"[1]. The integrative properties of multisensory perception are currently best explained in terms of optimal decision making using the Bayesian cue combination framework[2–7]. Ernst & Banks[8] first demonstrated that multisensory integration tends to be statistically optimal, in that it exploits all available sensory information, including redundant multisensory cues, to maximize the precision of sensory estimates. However, not all sensory signals should be integrated as not all simultaneous sensory inputs correspond to the same source in the distal world. Hence, a condition for optimal multisensory integration is to solve the "correspondence problem"[9], that is, detect redundancies in the continuous flow of multisensory inputs.

Given that information (i) is transduced by separate senses, (ii) converted in heterogeneous neural codes, and (iii) conveyed with variable latencies and hierarchical processes: how does the brain infer which information should be selectively integrated?

Auditory and visual signals that contain redundant information are typically generated by a common underlying physical cause and as such, they tend to correlate in time and space[10–16]. Accordingly, a large body of literature demonstrates improved multisensory integration when the constituent unimodal signals correlate in time and space[17–23]. These findings point to spatio-temporal correlation as the primary cue for solving the correspondence problem. Indeed, an important body of neurophysiological work has thoroughly described the neural tuning properties in many (sub)cortical regions to be sensitive to the spatial and to the temporal coincidence of multisensory inputs[24]. Multisensory neurons provide well-suited implementations for a recent Multisensory Correlation Detector model[25]. The Multisensory Correlation Detector (Fig. 1) detects and selectively integrates correlated audiovisual signals through a set of temporal

filters and elementary operations. Its outputs, $MCD_{CORR}$ and $MCD_{LAG}$, depend on the temporal correlation and order of the incoming auditory and visual signals, respectively. While being compatible with Bayesian ideal observer models, the Multisensory Correlation Detector also quantitatively describes the dynamics of sensory information processing. Whereas Bayesian models are mostly static and provide a snapshot of the overall response[26,27], the Multisensory Correlation Detector is a dynamic model in which both inputs and outputs are time-varying signals. These properties, combined with its biologically plausible nature[28,29], make the Multisensory Correlation Detector better suited for testing against intrinsically dynamic neurophysiological responses.

To explore this, we developed a time-resolved encoding model approach combined with time-resolved non-invasive neuroimaging (magnetoencephalography, MEG) to test the neurophysiological plausibility that such algorithms are implemented in the human brain.

First, we replicated the original behavioral experiment of[25], and had participants observe sequences of auditory clicks and visual flashes with varying temporal structures. Participants were recorded non-invasively with MEG in two separate blocks (Supplementary Fig. 1) testing two tasks. In a *causality judgment task*, participants judged whether the audiovisual sequences originated from the same source. This task probed the correlation detection of the model whose output is represented by $MCD_{CORR}$. In the *temporal order judgment task*, participants reported which of the acoustic or visual events appeared first in the sequence to probe the output of the lag detector ($MCD_{LAG}$). Crucially, identical audiovisual sequences were tested in both tasks. This experimental design thus maintained a constant and identical flow of feedforward multisensory inputs in both tasks, while manipulating the endogenous task requirements.

The gain of our approach is that we do not solely test the behavioral outcomes of temporal order or causality judgments[30,31]; rather, we test the presumed computations implicated in resolving temporal order or causal inference. Additionally, because Multisensory Correlation Detector is temporally-resolved, we use temporally-resolved neuroimaging to dissociate the outcomes of correlation and lag computations. This is all the more important as seminal fMRI work aiming at dissociating order and simultaneity perceptions in humans failed to find selective differences in brain regions implicated in the two tasks[32]. This suggests that computational differences are unlikely to be found in the location of macroscopic brain networks but rather in the subtle dynamical properties of those networks. In our study, and for each individual sequence of audiovisual stimuli presented to the participants, the Multisensory Correlation Detector makes quantitative predictions for both behavioral and neurophysiological responses, which we systematically tested against empirical evidence.

Below, we first validate our behavioral protocol in light of participants' causal and temporal order judgments for the same set of audiovisual sequences. We then illustrate differences of brain activity in the two tasks through classic analyses of MEG signals and finally demonstrate algorithmic differences by applying a time-resolved encoding model approach.

### Results

**The Multisensory Correlation Detector explains the behavioral responses in temporal order and causal judgments, by capturing the influence of the temporal structure.** The probability of responding "same cause" in the causality judgment task and "visual first" in the temporal order judgment task was predicted based on the time-averaged $MCD_{CORR}$ output and $MCD_{LAG}$ using a generalized linear mixed-model[33] with a logistic link function. Importantly, stimuli were chosen prior to the experimental work to probe largely uncorrelated judgments of causality and temporal

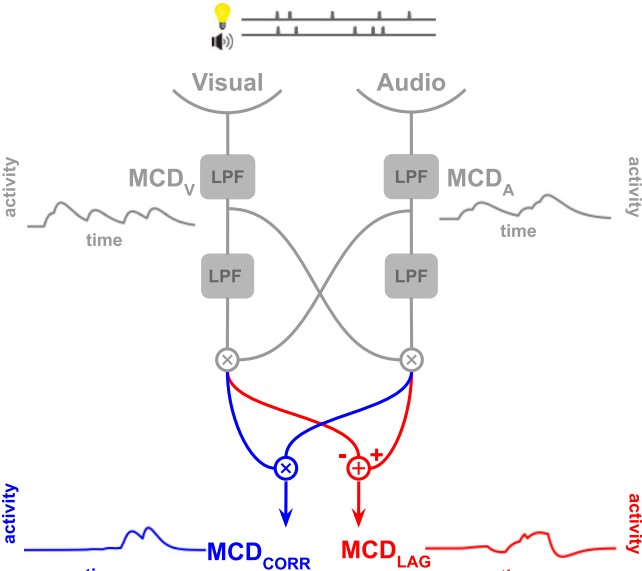

**Multisensory Correlation Detector (MCD)**

**Fig. 1 The Multisensory Correlation Detector is a set of low-pass filters (filled grey boxes), which compute the temporal correlation ($MCD_{CORR}$, blue) and the temporal lag ($MCD_{LAG}$, red) between incoming multisensory signals.** The model is biologically plausible as it integrates signals based on their cross-correlation in time and implements Bayesian optimal cue combination. LPF: low-pass filter.

order (see Methods). This was confirmed in the present results, as average causality and temporal order judgments were only weakly correlated across participants and sequences ($\beta = 0.267 \pm 0.08$, $p < 0.05$). The 3 parameters of the Multisensory Correlation Detector, controlling the time constant of the low-pass filters, were fitted using both tasks whereas the 2 parameters of the linear model were task dependent (see Methods). Results showed that the temporal structure of the stimuli systematically affected behavioural responses in both the causality and the temporal order judgment tasks (Fig. 2). $MCD_{CORR}$ activity was a good predictor of the causality judgments ($\beta = 1.80 \pm 0.05$, $p < 10^{-15}$, $R^2 = 0.92$). The positive $\beta$ indicates an increased probability of responding "same cause" when $MCD_{CORR}$ increases. Similarly, $MCD_{LAG}$ significantly predicted the temporal order judgments ($\beta = 1.02 \pm 0.04$, $p < 10^{-15}$, $R^2 = 0.81$). Interestingly, $MCD_{CORR}$ was a better predictor for the behavioural causality judgments than $MCD_{LAG}$ for the behavioural temporal order judgments (linear model comparing $R^2$ in both tasks, $\beta = 0.108 \pm 0.05$, $p < 0.05$, see individual fits in Supplementary Fig. 3). This suggests a possible asymmetrical dependency between causal inference and temporal order.

**Evoked responses elicited by causal judgments predict behavioral outcomes on a single-trial basis.** We hypothesized that if causality and temporal order judgments were mediated by multisensory correlation detectors, the same audiovisual sequences should elicit distinct patterns of brain activity corresponding to the predicted outcomes of the $MCD_{CORR}$ and $MCD_{LAG}$ computations, namely temporal correlation (needed for causality judgment) and lag calculation (needed for temporal order judgment), respectively. To test this, we contrasted brain activity obtained in response to the presentation of all audiovisual sequences in causality and temporal order judgment blocks. This contrast is informative for task-related brain activity: since identical audiovisual sequences were used in causality judgment blocks and in temporal order judgment blocks, feed-forward inputs were identical in both tasks. Hence, any differences observed in this contrast will reveal algorithmic differences imposed by the task.

In this contrast, we wanted to capture multisensory-specific activity. However, brain responses to multisensory events contain both unisensory-specific and multisensory-specific signals[34,35], which causes a superposition problem at the sensor levels. To solve this issue, we isolated the brain responses evoked by the multisensory operations from the responses elicited by unisensory stimulation: we removed the weighted sum of brain activity independently recorded in passive unisensory localizers (presentation of auditory alone or visual alone sequences) from the multisensory evoked activity recorded in causality and temporal order judgment blocks. This approach is compatible with heuristics used in fMRI[36] in that the weights were fitted with a linear regression (see Methods). We then contrasted the unisensory-free, *i.e.* multisensory-specific, evoked responses elicited by the same audiovisual sequences in the two tasks.

Statistical significance between conditions was assessed by spatiotemporal cluster permutation, controlling for multiple statistical comparisons at multiple latencies and samples (see Methods). This analysis revealed two bilateral significant clusters (the first is reported in Fig. 3; the second in Supplementary Fig. 4, individual values in Supplementary Table 3). The first bilateral cluster peaked around 700 ms following the onset of the audiovisual sequence ([260–1250 ms] in the left sensors; [540–1025 ms] in the right sensors, Fig. 3A). During this time window, the amplitude of the evoked response was significantly higher for causality than for temporal order judgments. This result was robust to two possible biases: first, the unisensory-free data may be biased by the weighting process and second, the lack of randomization of the response choice (fixed across blocks and participants) may contribute to the effect. We contend that this non-randomization could have introduced a motor response and/ or decisional bias in our results. We demonstrate the persistence of this significant cluster of activity without removing the unisensory-specific activity (see Supplementary Fig. 5) and when

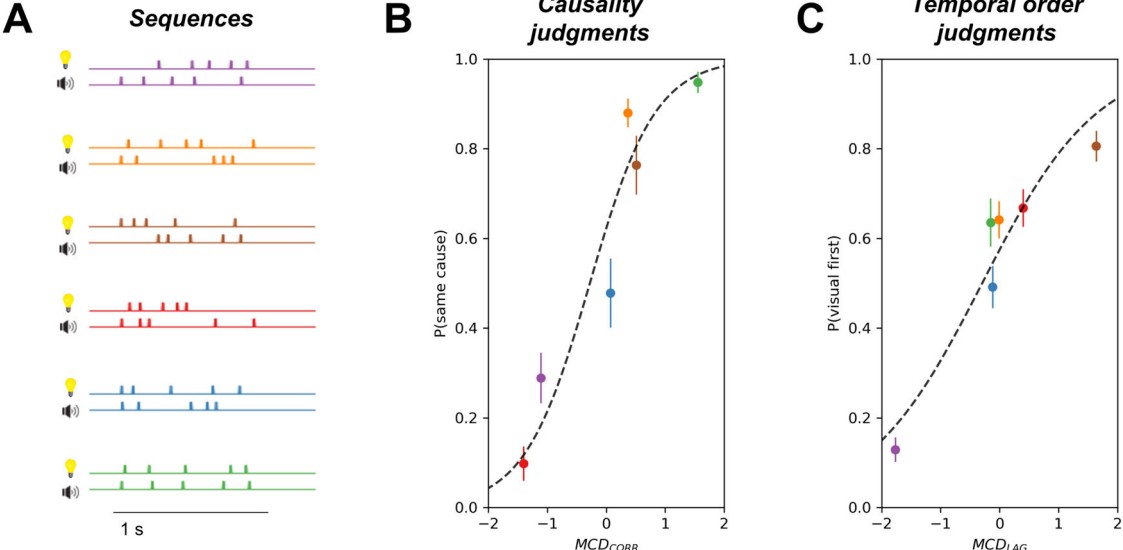

**Fig. 2 Psychophysical results. A** Audiovisual sequences. Stimuli were sequences of five clicks and flashes whose temporal statistics were manipulated so as to optimize the model predictions for the temporal correlation and for the temporal lag (see Methods). **B** The $MCD_{CORR}$ activity significantly predicts the causality judgments in the causality judgment task (mixed-effect logistic regression, $p < 10^{-15}$). **C** The $MCD_{LAG}$ activity significantly predicts the temporal order judgments in the temporal order judgment task (mixed-effect logistic regression, $p < 10^{-15}$). Each point corresponds to a different audiovisual stimulus. Given that the same six audiovisual stimuli were used for both tasks, points with the same colors in the left and right panels represent the participants' responses to the same stimuli in the two tasks. Error bars represent 2 s.e.m. (standard error of the mean, $N = 13$). Note that values of $MCD_{CORR}$ and $MCD_{LAG}$ differ from participant to participant because of the fitting procedure, but the error bars are smaller than the size of the points.

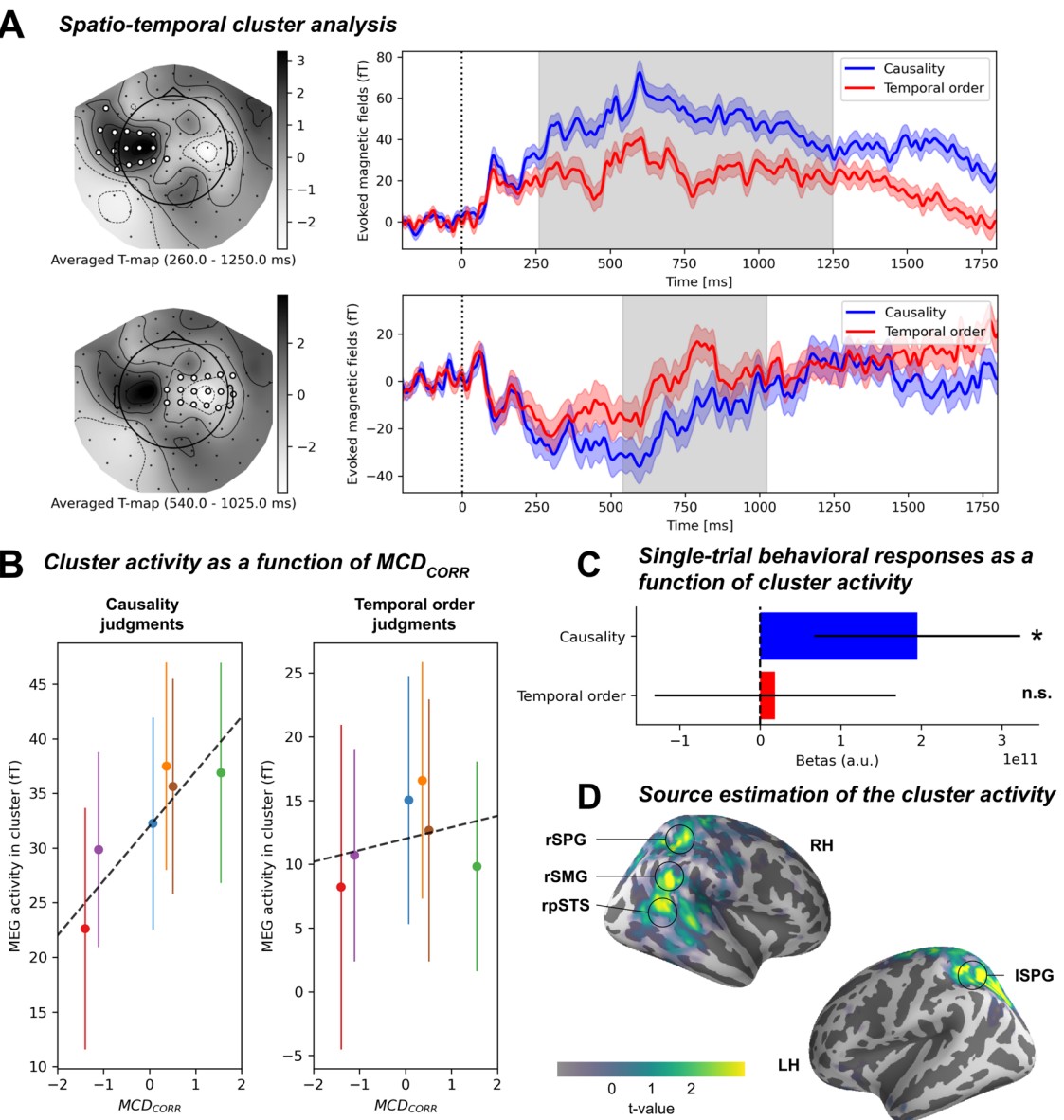

**Fig. 3 Task related evoked activity and cortical source estimates. A**. Spatiotemporal-cluster analysis contrasting brain activity evoked by the presentation of identical audiovisual sequences but in two different tasks (causality - temporal order). Left: t-map of the significant cluster (white sensors, one sample two-sided t-test, $p < 0.05$ corrected for multiple comparison) ranging from 260 to 1250 ms post-sequence onset. Grey levels are t-values averaged across significant times. Right: Temporal extent of the effect averaged over significant sensors in the cluster (grey). The top and bottom panels represent the two polarities of a single source (positive left, negative right) **B** Left: root mean squared (RMS) activity in the cluster average across participants as a function of $MCD_{CORR}$ in causality judgment blocks. Right: same in temporal order judgment blocks ($N = 13$). **C** Single-trial logistic regression coefficients showed that MEG activity can predict single-trial behavioral responses in causality judgment blocks (blue, $N = 13$, logistic regression $p < 0.05$) but not in temporal order judgment blocks (red, $N = 13$, logistic regression $p = 0.90$) **D** Source estimations (semi-inflated brain, $N = 13$) locate the effect to the right posterior superior temporal sulcus (pSTS), the right SupraMarginal Gyrus (SMG) and bilateral superior parietal gyrus (SPG). Shaded areas and error bars represent 2 s.e.m. across participants.

decomposing the task contrast analysis by response choice (see Supplementary Fig. 6).

A linear regression model (Fig. 3B, C) revealed that this evoked response was significantly correlated with $MCD_{CORR}$ ($\beta = 9.66 \pm 4.03$ fT, $p < 0.05$) in causality but not in temporal order judgments ($\beta = 2.37 \pm 3.76$ fT, $p = 0.53$, interaction $p < 0.05$). The positive $\beta$ indicates an increased evoked activity when $MCD_{CORR}$ increased. The regression with $MCD_{LAG}$ was not significant ($p > 0.05$).

To assess the behavioral relevance of this activity, we performed a single-trial logistic regression on the probability of participants' responses given the activity in the cluster (Fig. 3D).

A statistical model with the multisensory evoked activity, the task (causality *vs.* temporal order), and the interaction between both, outperformed all simpler models (see Methods). Given the significant interaction ($\beta = -0.32 \pm 0.159$ pT, $p < 0.05$), we looked at the main effect of evoked activity in both tasks. We found that the evoked activity in this cluster could predict single-trial responses in the causal inference task ($\beta = 195 \pm 128$ fT, $p < 0.05$) but could not predict participants' temporal order in the temporal order judgment task ($\beta = 19.0 \pm 15.2$ fT, $p = 0.90$).

Therefore, we have identified a brain response, which provides significant predictive information regarding the temporal correlation of audiovisual signals on a trial-by-trial basis, and which

predicts participants' causal inference. Source estimation was obtained by applying the inverse operator on the t-values values for each individual, and then morphing them onto the average brain. Source estimation (Fig. 3E) located this effect in the right posterior superior temporal sulcus, right supramarginal gyrus, and bilateral superior parietal gyrus. As predicted by the Multisensory Correlation Detector, at least one unit may compute the temporal correlation ($MCD_{CORR}$) between multisensory signals, and drive the responses of participants. In our study, this unit may be found bilaterally at the junction of the temporal and parietal cortices. The second significant cluster of task-related differences of activity showed no link with the overt reports (all $p > 0.9$) but did reveal differences in RTs linked to the motor responses (details are provided in Supplementary Fig. 4). Overall, we found no robust evidence for a lag unit ($MCD_{LAG}$) which would provide order information.

**The dynamics of the $MCD_{CORR}$ and the $MCD_{LAG}$ units correlates with the time-resolved brain activity**. The lack of robust evidence for the $MCD_{LAG}$ could arise from several possible caveats in our analysis. For instance, we estimated evoked activity integrated over time, and not the transient responses evoked by the fine temporal structure of the sequences. Additionally, evoked activity reflects phase-locked changes in brain responses that are strong and very reliable in time. Here, on the contrary, the stimuli have different temporal structures, which add substantial time variability in the evoked activity. Furthermore, the computation of the temporal cross-correlation during the one-second sequence entailed several fast iterations, following each sensory event. In other words, the integrated accumulated evidence of temporal jitters across audiovisual events, more than the single events, were crucial for performing in these two tasks[16]. Finally, in the absence of reference to the precise timing of the electrophysiological response, the correlations between the behavioral responses in causality judgments and the amplitude of the brain responses could conflate temporal correlation and late decisional processes that are irrelevant to the computations themselves, such as motor preparation (although participants were asked to withhold their answers after each stimulus presentation and until an auditory cue prompted them to answer).

To address these main concerns, we developed a novel model-based approach (Fig. 4A) consisting in using the dynamics of the Multisensory Correlation Detector units (audio and visual inputs, $MCD_{CORR}$, and $MCD_{LAG}$ outputs) in response to the audiovisual sequences presented in both tasks to explain MEG activity. Specifically, encoding models have been successfully applied to EEG and MEG recordings to understand speech processing[37–41]. The basic idea is that brain activity recorded with temporally-resolved neuroimaging can be decomposed as the convolution of the stimulus inputs and a filter. The filter, also known as "temporal response function", is usually unknown, and least-square fitted with a linear model. The quality of the resulting encoding model is then assessed with a cross-validation procedure with the model being fitted on one part of the data and evaluated on the unseen part. The cross-validated correlation between the predicted M/EEG signal based on the stimulus convolved with the filter and the actual data is a measure of the accuracy of the model. It allows producing a quantitative estimate of how much variance in brain activity can be explained by the characteristics of the stimulus.

While temporal response functions have essentially been used to assess the encoding of stimulus properties in brain activity, here we adapted this approach to test the encoding of the Multisensory Correlation Detector model itself. Hence, the cross-validated correlation gives a measure of brain activity that

can be explained, not just by the stimuli per se, but rather, by the Multisensory Correlation Detector model itself. Specifically, we used the modeled time-varying Multisensory Correlation Detector input and output signals in response to the stimuli. We refer to this method as "model-based temporal response function" (see Methods). It should be noted that this approach is related to, but distinct from, model-based approaches in fMRI. Here, we do not evaluate the correlation between a parameter value and a sensor/voxel across trials but directly the correlation between the dynamics predicted by the model and the dynamics of the MEG response. This aspect necessitates a time-resolved model, which is the case of the Multisensory Correlation Detector. Using the model-based temporal response functions, the cross-validated correlation gives a measure of the amount of data that can be predicted on the basis of the Multisensory Correlation Detector model. In other words, we used cross-validation correlations as a proxy of how much brain activity looked like the predicted Multisensory Correlation Detector signals. As a temporal response function can be fitted for each component of the Multisensory Correlation Detector, the model was resolved on a per component basis.

We first applied the model-based temporal response function method to the passive unisensory localizer responses as a proof of concept. In the auditory-only blocks, a temporal response function that took the auditory input component of the Multisensory Correlation Detector model ($MCD_A$) as predictor was able to predict the MEG activity ($R^2 \approx 2.2\%$). Cluster-based permutations revealed that the sensors best explained by the $MCD_A$ form a bilateral central cluster (Pearson's $\rho = 0.15$, $p < 0.001$, Fig. 4B). Source estimations (Fig. 4C) was obtained by applying the inverse operator on the correlation values for each individual, and then morphing onto the average brain. It located this pattern in bilateral posterior superior temporal gyri, consistent with auditory cortical sources. When applied to the visual-only responses, a temporal response function that took the visual input components of the Multisensory Correlation Detector ($MCD_V$) as predictor was able to predict the observed MEG activity ($\rho = 0.11$, $p < 0.001$). Cluster-based permutations revealed a significant bilateral posterior cluster of predicted sensors. Source estimations located the pattern in bilateral occipital cortices. These results validate the use of the temporal response function methodology to capture basic auditory and visual evoked responses, as well as their implementation in the Multisensory Correlation Detector.

Having ensured that the model-based temporal response functions provided coherent results for the unisensory localizers, we tested the experimental data on the causality and temporal order judgments. This time, instead of using only one predictor ($MCD_A$ or $MCD_V$), we used all four inputs/outputs of the Multisensory Correlation Detector. After fitting the temporal response functions, we set all but one predictor to zero to estimate the variance explained by each Multisensory Correlation Detector unit separately.

In causality judgment blocks, three units explained a significant part of the variance of the MEG activity (Fig. 4B): $MCD_A$ and $MCD_V$ predict very similar patterns as those observed in the localizer data, i.e. a bilateral central cluster ($\rho = 0.14$, $p < 0.001$), consistent with an auditory evoked response topography for $MCD_A$ and a bilateral posterior component consistent with a visual evoked response topography for $MCD_V$ ($\rho = 0.18$, $p < 0.001$). Critically, $MCD_{CORR}$ significantly predicted the MEG response in a bilateral central cluster ($\rho = 0.18$, $p < 0.001$), indicating that at least one component in MEG activity shows similar fluctuations as those predicted by the Multisensory Correlation Detector output. By contrast, $MCD_{LAG}$ had no significant predictive power ($\rho = 0.01$, $p = 0.1$) on the MEG

**A** *Model-based temporal response functions*

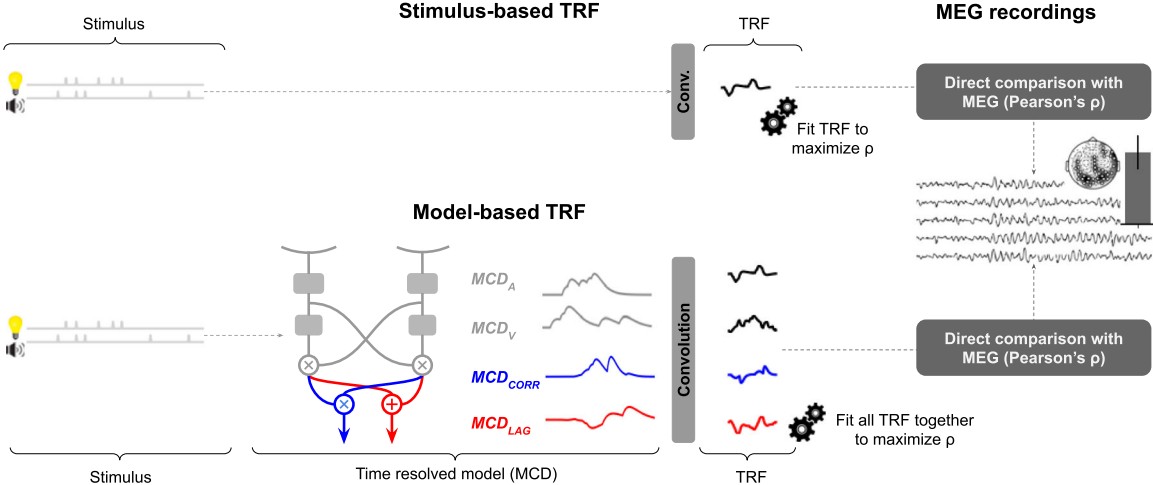

**B** *Explained variance per model unit, blocks and MEG sensors*

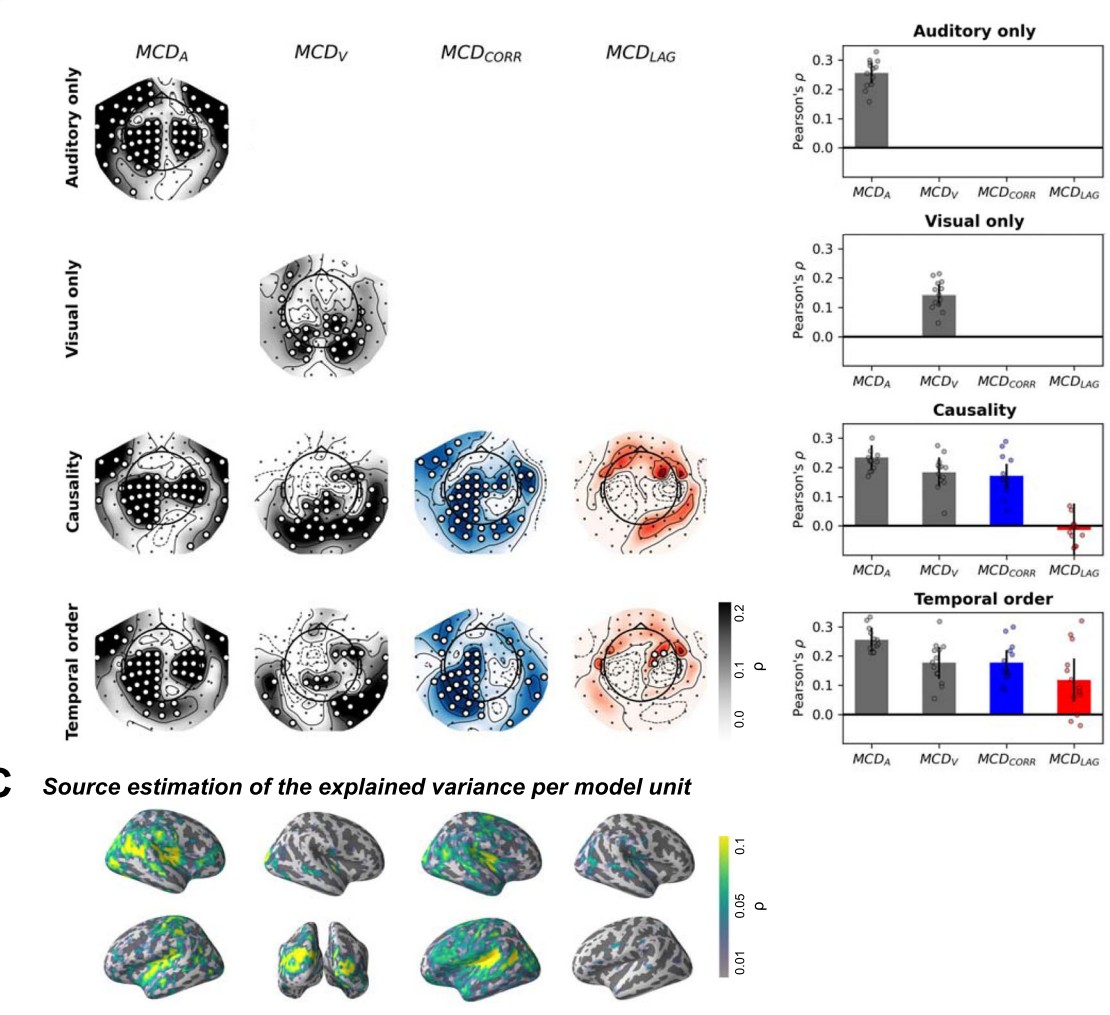

**C** *Source estimation of the explained variance per model unit*

activity. This result was expected for causality judgment blocks, in which participants solely evaluated the common cause of the audiovisual sequences irrespective of their temporal order.

When we applied the same approach to the temporal order judgment data, the temporal response function analysis provided a very similar pattern of results: as expected, the $MCD_A$ component predicted the activity in a bilateral central cluster ($\rho = 0.13$, $p < 0.001$), and the $MCD_V$ component in a bilateral posterior component ($\rho = 0.16$, $p < 0.001$). Critically, $MCD_{LAG}$ significantly predicted the MEG response in a left frontal cluster

**Fig. 4 Stimulus- vs. Model-based temporal response functions. A** Analysis pipeline. In the classic stimulus-based temporal response function approach (top), a linear filter estimates the amount of variance explained by stimuli in the MEG activity. In our model-based temporal response function approach (bottom), the filter is used to estimate the amount of variance explained by the model dynamics. This approach requires a time-resolved, biologically plausible, model, which is the case of the Multisensory Correlation Detector. The method estimates the amount of MEG responses variance explained by each component of the Multisensory Correlation Detector. For each component, a temporal response function is fitted by ridge regression such that the convolution between single trial component activity and the temporal response function maximizes the correlation between the predicted MEG signal and the true MEG signal. In order to avoid over-fitting, the ridge regression quality is evaluated through cross-validation. **B** The amount of MEG explained variance is resolved in model components, sensors, and tasks. The table reads as follows: in the causality judgment blocks (third row), the $MCD_A$ input component (first column) explains a significant proportion of variance in a bilateral central cluster, with a Pearson's $\rho = 0.16$ (right). Overall, $MCD_A$, $MCD_V$ and $MCD_{CORR}$ explain MEG variance consistently across causality and temporal order judgment blocks. To the contrary, $MCD_{LAG}$ significantly fit the data only in the temporal order judgment blocks. Statistical significance of the correlation between cross-validated predicted MEG and true MEG was assessed via corrected cluster permutations. Significant sensors are highlighted in white. Error bars represent 2 s.e.m. across participants (N=13). Dots represent individual participants. **C** Source estimations (inflated brain) locate $MCD_A$ in bilateral posterior superior temporal gyrus, $MCD_V$ in bilateral occipital cortices, $MCD_{CORR}$ in bilateral superior temporal gyrus, bilateral posterior superior temporal sulcus and bilateral supramarginal gyrus. $MCD_{LAG}$ is not associated with any robust phase-locked activation.

($\rho = 0.12$, $p < 0.005$), indicating that one component in MEG activity shows similar fluctuations as those predicted by the Multisensory Correlation Detector output. Source reconstruction could not reveal any consistent source. Interestingly and unexpectedly, despite participants engaging in a temporal order task, the $MCD_{CORR}$ showed a significant bilateral central cluster ($\rho = 0.17$, $p < 0.001$).

As the stimuli were not single events but sequences of events, later activity could, at least partly, be related to decisional processes based on the initial part of the sequence and not entirely related to the temporal computations themselves. To rule out this hypothesis, we replicated the analysis by splitting the evoked response in two halves: an early part, from 0 to 500 ms, and a late part, from 500 to 1000 ms (see Supplementary Fig. 7). The results show that the temporal response functions are stable across the two halves. This rules out the possibility that this activity reflects decisional processes, as decisional processes are supposed to happen only late in the sequence presentation. One further limitation could be that 4 out of 6 stimuli are perceived close to the 50/50 point in the temporal order judgment blocks. It could be the case that the ambiguity of these stimuli for this task is so great that the signals being measured in the brain are uncertainty plus guess, rather than temporal order computations per se. To rule out this hypothesis, we replicated the analysis in a reduced dataset comprising only the stimuli that are ambiguous in the temporal order judgment task (see Supplementary Fig. 8). The results show that the temporal response functions remain stable in this reduced dataset and that the results remain unchanged compared to the complete dataset. This rules out the possibility that the signal being recorded reflects only uncertainty or guesses.

Overall, the model based temporal response function we developed allowed the identification of a main set of brain signals, source localized in superior temporal sulcus, posterior superior temporal gyrus and supramarginal gyrus. All regions seemingly compute the temporal correlation between multisensory signals, independently of the task in our study. Source estimation also revealed a frontal right cluster of sensors, dedicated to the computations of the lag between multisensory signals, activated only when participants were explicitly performing a temporal order judgment task.

## Discussion

In this MEG experiment, participants made inferences about audiovisual sequences using two different judgments: common cause and temporal order. Behavioral results replicated prior observations[25], in that performances followed the predictions of the Multisensory Correlation Detector model. The temporal structure of audiovisual sequences is taken into account by participants in that the strength of audiovisual temporal correlation drives the $MCD_{CORR}$ activity, in turn affecting causal inference judgments. Similarly, the time lags between multisensory signals, which drive the $MCD_{LAG}$ activity also drive the perception of temporal order (Fig. 2). Second, the analysis of the MEG results corroborates several working hypotheses regarding the need to compute temporal correlation between multisensory signals. We found that MEG responses elicited by the presentation of the sequences (Fig. 3A) strongly correlate with behavior in causality judgments, but not in temporal order judgments (Fig. 3B, C). The likely generators of this activity were estimated in posterior superior temporal sulcus, superior temporal gyrus and left superior parietal gyrus (Fig. 3D), regions that are consistent with previous results in fMRI using a similar task[32] and similar stimuli[30]. Third, our model-based temporal response function analyses (Fig. 4A) strongly confirmed this result, showing that this activity is essentially driven by the $MCD_{CORR}$ output independently of the task. Furthermore, it allowed revealing a previously unknown activity, correlated to the $MCD_{LAG}$ output in the temporal order judgments (Fig. 4B).

Our results reveal that the Multisensory Correlation Detector model offers a multiscale explanatory power. The Multisensory Correlation Detector was originally conceived to provide a biologically plausible neural circuit for simultaneity and order detection of multisensory events. The few neurons circuitry has previously been described in the fly optic lobe[28]. Despite a massive scale change—from a few neurons to macroscale synchronized neural populations –, we were able to find evoked activity, which matched the Multisensory Correlation Detector computation dynamics. This model thus offers a good fit to the behavioral data and to human neural activity.

The achievement of a model-based temporal response function for human MEG data was only made possible because the Multisensory Correlation Detector provides time-resolved biologically plausible signals. One prerogative of classical model-based approaches—including fMRI—is to artificially select discrete values of the model parameters on a trial-by-trial basis. Here, we could perform on human non-invasive neurophysiological data what can typically only be done in animal models, namely directly compare the dynamics of the model computations and the neural signals on a single-trial basis. This ensures that the signals we capture are not related to late decisional processes, but to the computations themselves. Thus, and according to the conceptual definitions of David Marr[42,43], the type of explanation we provide here is of an implementational nature of posited computations, and not solely one based on an algorithmic explanation given by a model-based approach.

Finally, the model-based temporal response function approach differs from classical temporal response function, which typically compares brain responses to speech stimuli[38,44]: while temporal response functions empirically fit the data to recover the canonical response function of the sensory inputs from continuous brain recordings, here, we provide an explicit model of the computation that actually generates the data and which we then try to recover in the brain signals. It should also be noted that the model-based approach suffers less from the influence of stimulus statistics specificities, such as the presence of strong onsets[45].

Interestingly, we found possible traces of the $MCD_{CORR}$ computations independently of whether participants were performing the causality or the temporal order judgment. This finding suggests that the computation of the temporal correlation between signals may be a default operation present in both tasks as predicted by Multisensory Correlation Detector and that this computation may be automatic, *i.e.* independent of specific task demands. The possibility that multisensory integration may be a default integrative mode for the representation of sensory signals supports recent findings[46] and is consistent with perceptual integration of multisensory dynamic structures[19,23] and speech processing[13,14]. To the contrary, a reliable fit to the $MCD_{LAG}$ could only be observed in the temporal order judgment task, suggesting that reliable multisensory segregation may be more effortful[47] or modulated by nonlinear network dynamics[48,49]. However, we cannot strongly conclude for the co-existence of correlated and lag signals, or for the absence of lag signals in the causality judgment because of methodological constraints due the presence of correlation between the $MCD_{CORR}$ and other subunits of the Multisensory Correlation Detector. Furthermore, 4 out of 6 stimuli were perceived as ambiguous with regard to the temporal order judgment. Such a high level of uncertainty could have reduced the detectability of the $MCD_{LAG}$ effect on the MEG recordings.

A recent debate has taken place over whether multisensory causal inference recruits frontal regions or solely anterior parietal cortex[46,50–52]. The lateral prefrontal cortex has been reported multiple times as a key region for multisensory convergence and multisensory conflict resolution[53–56]. It is also a region suggested to play a key role in reverse hierarchical learning following multisensory training[19,23]. One discrepancy between previous results could be explained by differences in the method of analysis as most previous studies reporting strong frontal effects relied on multivariate analyses. For instance, one recent study[50] found the implication of the lateral prefrontal cortex in encoding causal decision in a causal inference task but only when relying on a multivariate decoding approach. Alternatively, as seen in the Multisensory Correlation Detector, cause and order result in two distinct signals, which may follow separate paths: as a result, while the output serving causal judgment ($MCD_{CORR}$) is readily captured by the model, the output serving temporal order ($MCD_{LAG}$) may require additional steps to yield behavioral decision. This observation is consistent with the fact that temporal order and simultaneity judgments do not readily correlate intra-individually[57]. It is also consistent with previous MEG work which, using both univariate and multivariate analyses, showed the implication of prefrontal cortices in the temporal aspects of multisensory integration[19,23]. Further, the implication of prefrontal cortices in temporal order judgements is also consistent with their known role in the temporal sequencing of events[58] and in decision-making tasks that involve a mixture of feedforward and feedback processes[59].

Overall, our work bridges three complementary approaches to understand how the human brain solves the correspondence problem and perform multisensory integration: (i) computational models, derived from animal electrophysiology, (ii) behavioral work in human, and (iii) non-invasive human electrophysiology.

The analytical tool we introduce is an important step to bridge animal and human work[60] and to test biophysically sound models across species.

## Methods

**Participant details**. Thirteen participants (10 females, mean age 25.8 y.o., range [22-34 y.o.]) took part in the study. All were right-handed, had normal hearing and normal or corrected-to-normal vision. Prior to the experiment, all participants provided a written informed consent in accordance with the Declaration of Helsinki (2008) and the local Ethics Committee on Human Research at NeuroSpin (Gif-sur-Yvette, France). The sample size was determined on the basis of the original paper describing the psychophysical task and main effects[25], ($N = 5$) and is comparable to studies using similar stimuli and paradigms (for example[46]). A sensitivity analysis is available in Supplementary Fig. 9. Additionally, we conducted post-hoc observational power analysis based on Monte-Carlo simulations[61] on our two main behavioral results, namely the effect of $MCD_{CORR}$ on causality judgments and the effect of $MCD_{LAG}$ on temporal order judgments. The simulations showed that the generalized linear modeling approach with our sample size and our observed effect sizes shows excellent power (95% CI [99.63, 100.0] and [99.80, 100.0]). There is currently no gold standard to evaluate sample size for MEG data, so we used the common sample size in the field.

**Multisensory correlation detector model**. The Multisensory Correlation Detector (Fig. 1) stems from the idea that the multisensory correspondence problem is computationally similar to the problem of visual motion detection (*are the two visual objects present at two different places at two different times or is it one single moving object?*). The proposal is that the computation of the cross-correlation in the time domain is the main mechanism underlying the perception of visual motion and auditory localization. Parise & Ernst (2016) therefore proposed an adaptation of the famous "Hassenstein-Reichardt detector" to the correspondence problem in multisensory perception. The Multisensory Correlation Detector is composed of two unisensory input units, a set of low-pass filters, and two multisensory output units. The low-pass filters smear and delay inputs from one sensory modality relative to the other, such that it computes the correlation across all possible delays. This operation approximates temporal cross-correlation. The two output units code for the two aspects of a temporal cross-correlation between the two input signals: the correlation unit ($MCD_{CORR}$) code for the strength of the correlation, and the lag unit ($MCD_{LAG}$) code for the sign and size of the lag. Formally, the Multisensory Correlation Detector is composed of a first filtering stage, where time-varying visual and auditory signals ($S_A(t)$, $S_V(t)$) are independently low-pass filtered, and a subsequent integration stage, where the two signals are combined through linear operations (multiplication or subtraction). Low-pass filters were modelled as exponential functions with $\tau_{mod}$ denoting the modality-dependent time constant and $*$ denoting the convolution operator:

$$f_{mod}(t) = t e^{\frac{-t}{\tau_{mod}}} \qquad (1)$$

$$MCD_A(t) = f_A(t) * S_A(t) \qquad (2)$$

$$MCD_V(t) = f_V(t) * S_V(t) \qquad (3)$$

Each subunit ($MCD_{S1}$, $MCD_{S2}$) of the detector independently combines multisensory information by multiplying the filtered visual and auditory signals as follows:

$$MCD_{S1}(t) = (MCD_A(t) * f_{AV}(t)) \cdot MCD_V(t) \qquad (4)$$

$$MCD_{S2}(t) = (MCD_V(t) * f_{AV}(t)) \cdot MCD_A(t) \qquad (5)$$

The response of the subunits are eventually multiplied or subtracted:

$$MCD_{CORR}(t) = MCD_{S1}(t) \cdot MCD_{S2}(t) \qquad (6)$$

$$MCD_{LAG}(t) = -MCD_{S1}(t) + MCD_{S2}(t) \qquad (7)$$

**Experimental design**. The main MEG experiment (Supplementary Fig. 1) consisted of 10 consecutive recording blocks of 8 min each, whose order was counterbalanced across participants. Three blocks tested participants on a causality judgment, and three blocks tested participants with a temporal order judgment. Importantly, the same audiovisual sequences were used in both tasks in order to maintain a constant flow of feedforward multisensory inputs while manipulating the endogenous task requirements. Each bloc was composed of 25 repetitions of the 6 possible audiovisual sequences. A total of 75 presentations of each stimulus sequence were thus tested in each task. Four additional recording blocks consisted of participants passively hearing (A localizer, 2 blocks) or viewing (V localizer, 2 blocks) one constitutive modality of the audiovisual sequence. Each localizer block was composed of 25 repetitions of the 6 possible stimuli (auditory or visual part of each stimuli), yielding a total of 50 presentations of each A and V stimuli ($2 \times 3 \times 25 \times 6 + 2 \times 25 \times 2 \times 6 = 1500$ trials

in total). The goal of localizer blocks was to model unisensory responses and source estimates that were orthogonal to the brain activity recorded during the task. Participants were asked to take breaks as often and as long as they wanted to between blocks. Prior to the MEG acquisition, participants were briefly familiarized with the task and the stimuli by performing three causality judgments and three temporal order judgments on audiovisual sequences which were not used in the experiment thereafter. Participants did so with the help of the experimenter and received feedback on their performance. No feedback was provided in the remainder of the experimental session. The experiment lasted for approximately 90 minutes.

In the causality judgment blocks, participants judged whether the auditory and the visual sequences were causally related or not, *i.e.* whether they were produced by the same common cause or not[25]. In the temporal order judgment blocks, participants judged which of the auditory or visual sensory modality was leading in time. In both causality and temporal order judgment blocks, participants responded via button presses: in the causality judgment blocks, they used the right index finger for "same cause", and the right middle finger for "different cause"; in the temporal order judgment blocks, they used the right index finger for "sound leading", and the right middle finger for "vision leading". In both causality and temporal order judgment blocks, trials consisted of audiovisual sequences separated by inter-trial time intervals pseudo-randomly selected between 500 and 900 ms. In a given trial, following the presentation of the sequence, participants were asked to withhold their answers until an auditory cue (10 ms, 1000 Hz pure tone) prompted them to answer. The time interval between the end of a stimulus and the auditory cue was randomly chosen between 800 and 1200 ms. The order of the stimuli was pseudo-randomized, so that none of the sequences were presented more than three times consecutively.

**Stimuli**. Stimuli were designed and presented using Matlab (R2012a, Mathworks Inc.) with Psychtoolbox-3[62] on a PC (Windows XP). Six audiovisual stimuli were used in the experiment and each consisted of a one-second sequence of 5 auditory bursts and 5 visual flashes (see Supplementary Table 1). Sounds were 10 ms bursts of pink noise, whose intensity linearly ramped-on for the first half of the time, and off for the second half. The sampling frequency used was 44.1 kHz. The sound pressure level was set to a comfortable hearing level, around 70 dB, for all participants. Visual events were 10 ms flashes, delivered by a white LED. To ensure spatial congruence, the LED was directly fixed on the speaker facing the participant. The LED and the speaker were controlled by the same sound card, allowing near-perfect timing and temporal congruence between sound and vision.

The temporal structure of the audiovisual sequences was a key factor in the experiment. The temporal structure was designed to make the causal and order judgments vary independently. Indeed, in order to study the unique contribution of neural sources to causality and temporal order judgments, it is critical to (1) use the same stimuli and (2) to have uncorrelated judgments across stimuli (*i.e.*, stimuli associated with a given causality judgment should be systematically associated with a given temporal order judgment). For that, we drew $10^6$ random audiovisual sequences and computed the time-averaged values of $MCD_{CORR}$ and $MCD_{LAG}$ for each sequence, with the parameters reported in[25]. We then selected six sequences among these $10^6$ to elicit a large range of responses of $MCD_{CORR}$ and $MCD_{LAG}$ values, while maintaining the correlation between $MCD_{CORR}$ and $MCD_{LAG}$ low (Pearson's correlation between $MCD_{CORR}$ and $MCD_{LAG}$ across the six sequences below 0.2). Pretests were run to ensure that causality and temporal order judgments were indeed uncorrelated. In the present experiment, the two judgments are weakly correlated across participants and sequences ($\beta = 0.267 \pm 0.08$, $p < 0.05$), thus allowing contrast analysis on the task.

**MEG data acquisition**. Brain magnetic fields were recorded in a magnetic-shielded room using a 306 MEG system (Neuromag Elekta LTD, Helsinki). Participants were seated in upright position under the MEG dewar, facing a white LED, mounted on a head-speaker placed 90 cm away. They were explained with the task and stayed in contact at all times with the experimenter via a microphone and a video camera. They were asked to refrain from blinking during the presentation of the audiovisual sequences. MEG recordings were sampled at 1 kHz and band-pass filtered between 0.03 Hz and 1 kHz. Four head position coils (HPI) measured the head position of participants before each block. Three fiducial markers (nasion and pre-auricular points) were used for digitization and anatomical MRI (aMRI) immediately following MEG acquisition. Electrooculograms (EOG, horizontal and vertical eye movements) and electrocardiogram (ECG) were simultaneously

recorded. Prior to the session, 2 minutes of empty room recordings were acquired for the computation of the noise covariance matrix.

**Anatomical MRI acquisition and segmentation**. The T1-weighted aMRI was recorded using a 3-T Siemens Trio MRI scanner. Parameters of the sequence were: voxel size: $1.0 \times 1.0 \times 1.1$ mm, acquisition time: 466 s, repetition time TR: 2300 ms and echo time TE: 2.98 ms. Cortical reconstruction and volumetric segmentation of participants' T1-weighted aMRI was performed with FreeSurfer (http://surfer.nmr.mgh.harvard.edu/). This includes: motion correction, average of multiple volumetric T1-weighted images, removal of non-brain tissue, automated Talairach transformation, intensity normalization, tessellation of the gray matter white matter boundary, automated topology correction and surface deformation following intensity gradients. Once cortical models were complete, deformable procedures could be performed including surface inflation and registration to a spherical atlas. These procedures were used with MNE_python[63] to morph current source estimates of each individual onto the FreeSurfer average brain for group analysis.

**MEG data preprocessing**. Data preprocessing was done in accordance with accepted guidelines for MEG research[64]. Signal Space Separation (SSS) was carried out using MaxFilter to remove external interferences and noisy sensors. Signal-space projections were computed by independent component analysis (ICA) to correct for eye blinks and cardiac artifacts. Next, raw data were band-pass filtered between 1 and 40 Hz and down-sampled to 250 Hz. All pre-processing steps were done using MNE_python 0.24.

**MRI-MEG co-registration and source reconstruction**. The coregistration of MEG data with the individual's structural MRI was carried out by realigning the digitized fiducial points with MRI slices. Using. MRILAB (Neuromag-Elekta LTD, Helsinki), fiducials were aligned manually with the multimodal markers visible on the MRI slice. An iterative procedure was subsequently used to realign all digitized points (about 30 more supplementary points) distributed on the scalp of the participants were digitized, with the scalp tessellation using MNE_python tools.

Individual forward solutions for all source reconstructions located on the cortical sheet were next computed using a 3-layers boundary element model[65,66] constrained by the individual aMRI. Cortical surfaces were extracted with FreeSurfer and decimated to about 10,240 vertices per hemisphere with 4.9 mm spacing. The forward solution, noise and source covariance matrices were used to calculate the depth-weighted (parameter gamma = 0.8) and noise-normalized dynamic statistical parametric mapping (dSPM)[67] inverse operator. This unitless inverse operator was applied using loose orientation constraints on individuals' brain data by setting the transverse component of the source covariance matrix to 0.4. The reconstructed current orientations were pooled by taking the norm, resulting in manipulating only positive values. The reconstructed dSPM estimates time series were morphed onto the FreeSurfer average brain for group analysis and common referencing.

**MEG data processing and evoked activity**. All preprocessed MEG data were epoched from $-200$ ms to $+1800$ ms around the onset of the first event in each sequence. Epochs contaminated by artifacts were rejected based on the peak-to-peak amplitude ($2.10^{-11}$ T for magnetometers and $9.10^{-9}$ T cm$^{-2}$ for gradiometers). Across participants, an average of 98.9% of epochs were kept in the analyses. Sensors were then interpolated using the function EvokedArray.as_type of MNE_python: each triplet of planar gradiometer, axial gradiometer and magnetometer was mapped onto a single virtual magnetometer, reducing the data space from 306 to 102 sensors without discounting signal or information. Epochs were averaged for each participant across experimental blocks and conditions of interest.

In order to focus on multisensory and task-related effects, we first removed unisensory evoked responses from the multisensory brain responses. In order to do this, we fitted a linear regression to the evoked responses elicited by audiovisual sequences in the causality and temporal order judgments blocks using, as predictors, the brain activity evoked by the unisensory (A,V) localizers. The formulae below summarizes this operation in which: $mMEG_{CJ}(n, t)$ designates the unisensory-free evoked activity in causality judgment blocks of the sensor n at time t; $mMEG_{TOJ}(n, t)$ designates the unisensory-free evoked activity in temporal order judgment blocks of the sensor n at time t; $MEG_{CJ}(n, t)$ the raw evoked activity in causality judgment blocks of the sensor n at time t;

**Table 1 Fitted parameters of the Multisensory Correlation Detector for each participant.**

| ID | 1 | 2 | 3 | 4 | 5 | 6 | 7 | 8 | 9 | 10 | 11 | 12 | 13 |
|---|---|---|---|---|---|---|---|---|---|---|---|---|---|
| $\tau_V$ (ms) | 130 | 137 | 120 | 152 | 142 | 141 | 122 | 123 | 114 | 109 | 103 | 119 | 122 |
| $\tau_A$ (ms) | 84 | 80 | 58 | 110 | 92 | 73 | 50 | 53 | 52 | 52 | 54 | 64 | 74 |
| $\tau_{AV}$ (ms) | 519 | 490 | 562 | 1147 | 882 | 1087 | 748 | 372 | 764 | 779 | 799 | 779 | 770 |

MEG$_{TOJ}$(n, t) the raw evoked activity in temporal order judgment blocks of the sensor n at time t; MEG$_A$(n, t) the raw evoked activity in the A block of the sensor n at time t; MEG$_V$(n, t) the raw evoked activity in the V blocks of the sensor n at time t. Parameters $\beta_A$(n) and $\beta_V$(n) were fitted by linear regression on a per sensor (n), per participant and per block across all time points and all stimuli.

$$mMEG_{CJ}(n, t) = MEG_{CJ}(n, t) - (\beta_A(n)MEG_A(n, t) + \beta_V(n)MEG_V(n, t)) \quad (8)$$

$$mMEG_{TOJ}(n, t) = MEG_{TOJ}(n, t) - (\beta_A(n)MEG_A(n, t) + \beta_V(n)MEG_V(n, t)) \quad (9)$$

### Statistics

*Multisensory Correlation Detector model fitting.* The analysis was mainly focused on behavioral ratings considering that reaction times (RTs) were delayed in the task due to participants having to wait for a prompt to deliver their answers.

We used the same fitting procedure as the original paper[25]. The probability of responding "same cause" in the causality judgment task was predicted using the MCD$_{CORR}$ values, and the probability of responding "visual first" in the temporal order judgment task was predicted using the MCD$_{LAG}$ values. The MCD$_{CORR}$ and MCD$_{LAG}$ values were time-averaged in a 3s-long time window starting from the first impulse—three times the duration of each sequence —, in order to have one value of each per trial. These values were then z-scored across stimuli to allow comparing more easily the effects of each on behavior and on MEG activity. The unnormalized values can be found in Supplementary Table 2. These time-averaged values were then entered as fixed effects in a mixed-effect logistic regression on the probability of responding "same cause" or "visual first", with a random intercept for each participant. There were 7 parameters to fit: 3 parameters for the Multisensory Correlation Detector model (the time constants of the low-pass filters), and 2 parameters for the cumulative gaussian for each task (controlling the intercept and the slope). The parameters were all fitted together: the 3 parameters of the Multisensory Correlation Detector were fitted using both tasks whereas the 2 parameters of the mixed-effect logistic regression and the random slopes per participant were task independent. The fit consisted of a particle swarm algorithm followed by Matlab optimization algorithm fminsearch to maximize model's log likelihood. Since both methods are sensitive to starting parameter values, the procedure was repeated 10 times and the chosen parameter values are those providing the best fit (Table 1). Reported p-values are Satterthwaite approximations obtained using R (R Core Team, 2000) with the lmerTest[68] and lme4[69] packages. Difference of goodness-of-fit between model units was assessed using a paired t-test on the $R^2$ across participants.

*MEG spatio-temporal clusters.* Evoked response fields (ERFs) in each task were tested by contrasting the unisensory-free activity for each audiovisual sequence in the causality and temporal order judgment blocks. The significance of the contrasts was assessed using non-parametric pairwise two-tailed permutation tests across time and sensors, which provided corrected p-values for multiple comparisons. The null hypothesis is that the MEG amplitude should be the same whatever the task. The statistic of interest in the t-value of the paired t-test contrasting the task across participants and stimuli. Spatio-temporal clusters were defined on the basis of temporal adjacency and covariance matrix by regrouping samples whose T-statistic was larger than 2.6, corresponding to a p-value ≤ 0.01 for a T-test with 77 degrees of freedom (13 participants x 6 sequences– 1). Cluster-level statistics were then calculated by taking the sum of the t-values within the cluster. We used random permutations to build the null distribution of the cluster-level statistics. On each permutation (N = 10 000), task labels were randomly shuffled across trials[70]. Only spatio-temporal clusters with permutation corrected p-values ≤ 0.01 are reported.

*MEG x behavior correlation.* The activity in each cluster was correlated with behavioral responses to assess how relevant this differential activity was to participants' response. We fitted generalized linear mixed-models on the probability of a response with a logistic link function. The root mean squared (RMS) evoked activity in the cluster was set as a fixed effect, and the participant was set as the random effect. All models were fitted on single trial evoked activity per participant.

### Model-based temporal response functions.

Temporal response functions are encoding models of MEG activity[37] and rely on the assumption that MEG activity can be expressed as a linear convolution between the input stimuli and a filter. The filter is typically unknown and therefore estimated by a least-square ridge regression. We extended this method by using the Multisensory Correlation Detector model signals as input instead of the stimuli. We refer to this method as "model-based temporal response function". Technically, the relation between the Multisensory Correlation Detector signals and the MEG is modelled as follows:

$$MEG(n, t) = \sum_i \sum_{\tau=t_{min}}^{t_{max}} TRF(n, i, \tau) \cdot MCD(i, t - \tau) + \varepsilon(n, t) \quad (10)$$

Where: MEG($n$, $t$) is the MEG activity at of the sensor $n$ at time $t$, TRF($n$, $i$, $\tau$) is the value of the temporal response function for the sensor $n$, the Multisensory Correlation Detector unit $i$, at lag $\tau$, MCD($i$, $t$-$\tau$) is the value of the Multisensory Correlation Detector unit $i$ at time $t$-$\tau$, $\varepsilon(n, t)$ is the residual error at time $t$ for the MEG sensor $n$.

The residual error $\varepsilon$(n, t) is minimized for each sensor n using least-square regression. Note that we fit one temporal response function per MEG sensor and per Multisensory Correlation Detector component. The parameters of the Multisensory Correlation Detector model were estimated from the behavioral data and left untouched at this stage of the analysis. Overfitting was handled by using a standard procedure of cross-validation: at each iteration, we fitted temporal response functions on all participants and all stimuli except for one stimulus of one participant (78-folds cross-validation). We evaluated the performance of the temporal response function for each component of the Multisensory Correlation Detector separately by Pearson's ρ between predicted activity and real MEG recordings. Finally, ρ scores were averaged across all folds and tested against 0 with a two-tailed t-test. The output of this test was corrected for multiple comparison using spatial cluster permutations (N = 10 000, threshold T = 2.6).

**Reporting summary.** Further information on research design is available in the Nature Research Reporting Summary linked to this article.

## Data availability

The data that support the findings of this study have been anonymized, defaced and converted to the BIDS format[71] using MNE-BIDS 0.8[72]. Source data are provided with this paper. The behavioral and MEG data generated in this study and the custom codes used to analyze the data have been deposited in the OpenNeuro database under accession code https://openneuro.org/datasets/ds003922.

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

## Acknowledgements

We thank the members of UNIACT for their logistical help on the recruitment of volunteers at NeuroSpin. We thank Leila Azizi for her help with the LED apparatus. This work was supported by an ERC-YStG-263584 and an ANR-16-CE37-0004-04 to V.v.W.

## Author contributions

Conceptualization J.P.L. and V.v.W.; Data curation J.P.L.; Formal Analysis C.V.P. and J.P.L.; Funding acquisition V.v.W.; Investigation J.P.L.; Methodology J.P.L. and V.v.W.; Project administration V.v.W.; Resources M.O.E. and V.v.W.; Software V.v.W.; Supervision M.O.E. and V.v.W.; Validation M.O.E. and V.v.W.; Visualization J.P.L.; Writing –

original draft J.P.L. and V.v.W.; Writing – review & editing J.P.L., C.V.P., M.O.E. and V.v.W.

## Competing interests

The authors declare no competing interests.
