## [Peer Review File · Nature Communications]

Multisensory correlation computations in the human brain identified by a time-resolved encoding modelREVIEWER COMMENTS

Reviewer #1 (Remarks to the Author):

Capitalizing upon an elegant computational model (Parise & Ernst, 2016), the authors of this manuscript sought to find neural correlates of “input-driven” model-based multisensory representations in the human brain. The behaviour of their human subjects can be well described by the multisensory correlation detector (MCD), replicating previous findings. Contrasting magnetoencephalographic data in different cognitive tasks on identical sensory inputs, the authors find task-related neural effects in and around parietal-temporal brain regions. A dynamic encoding model further reveals unisensory representations in respective unisensory cortices, neural representation of a correlation detector in parietal-temporal regions, and a temporal-order detector in frontal area. I think this manuscript makes a potential contribution to our understanding of how the physiologically-plausible MCD might actually be linked to the brain, thus extending that 2016 modelling paper. The dynamic encoding modelling is innovative and particularly suitable for addressing research questions about spatiotemporal dynamics in MEG data. However, I have several concerns pertaining to their analysis methods and data interpretation.

Major concerns:

1) The MCD model parameters, i.e., 3 time constants, were estimated by fitting the model to both tasks at the same time. I find this decision unwarranted. Why are the time constants a priori assumed to be unchanged across tasks? Fig. 2 shows that the psychometric functions differ between the two tasks. I worry a changing time constant(s) across the tasks would lead to similar results. Here is a suggestion on how to justify your choice of the fixed model parameters across tasks: instead of fitting the model to your data by maximising the correlation between empirical vs. model-predicted psychophysical kernel, which relies on collapsing trials, you can fit the model to trial-by-trial behavioural responses by maximising binomial log likelihood. You can include a noise parameter for the transfer function linking MCD to choice e.g., a softmax or cumulative Gaussian, on a trial-by-trial basis. You can fit this model (4 free parameters: 3 time constants + 1 noise parameter) to each task independently, and see if any parameter differs systematically across tasks. It’s been well-known that, to learn and make decisions appropriately, humans and other animals have evolved to integrate various sensory information on multiple timescales. These time constants are essentially geared to capture such flexible timescales in the model and are likely to change as a function of cognitive tasks. This concern affects greatly the interpretation of their neural results.

2) Motor-activity/response-bias confound. Across the two tasks, the response-choice mapping was fixed in all blocks for all subjects. In causality judgment, subjects used the right index finger for “same cause” choice and the right middle finger for “different cause”; in temporal-order judgment, they used the right

index finger for “sound leading” choice and the right middle finger for “vision leading”. This design would lead to two potential confounds when interpreting the MEG results in Fig. 3. Firstly, the neural effect of task change may reflect change in motor effect or response bias. It is very possible that, for the same stimulus, a subject used the index finger to commit a “same cause” choice in one task and the middle finger for a “vision leading” choice in the other task. Simple solutions such as 1) including binary finger code as covariates in the task contrast analysis or 2) running the task contrast only for trials with identical finger response would rule out this confound. But the data in Fig. 3A so far can’t be interpreted as a genuine neural effect of task change, above and beyond motor-related signal change. Of note, those significant ‘white sensors’ in Fig. 3A (also in Supp Fig. 4A) are mostly in the left hemisphere, which may be echoing this motor confound. Secondly, the effect of model-predicted MCD_corr on MEG activity (Fig. 3B left) may reflect motor effect because MCD_corr is a strong predictor for choice (= finger). The striking similarity between the wavy pattern in Fig. 3B left and that in Fig. 2B indicates that the MEG activities may indeed reflect finger identity.

3) Another concern is about regressing out unisensory signals from multisensory signals. Of note, the authors did this regression separately for each task. This means that the residuals in the multisensory signal are those residual activities that cannot be explained by any weighted combination of the unisensory auditory and visual signals alone. However, it has been known that change of cross-modal decision strategies can be reflected by the change of linear neural weight assigned to each modality (Rohe, Ehlis & Noppeney, 2019 Nat. Comm.). If one regresses out the flexible combination of unisensory signals from each task separately by allowing 4 free weights (2 for A/V in CJ task and 2 in TOJ task), the residuals in the multisensory condition would completely miss out the chance of describing any neural effect in the brain that potentially corresponds to this task-related weight change. Even though I understand that one may reasonably assume that only the ‘nonlinear’ effect (above and beyond weighted average) is interesting when speaking of multisensory interaction, I find this residualization approach unwarranted. For instance, the authors made a strong assumption that the brain signals evoked by unisensory inputs can be readily generalisable to multisensory conditions as ‘component’ signals at the exact MEG sensor and at the exact moment in time (this is how the residualization was done). But it is unclear if the spatiotemporal pattern may have been altered by change of attention etc. across unisensory vs. multisensory conditions, which makes that assumption a bit shaky.

4) MCD_corr(t) is the product of MCD_A(t) and MCD_V(t), whereas MCD_lag(t) is the linear combination of them. This in fact would encourage alternative interpretation of the results in Fig. 4. It is unclear whether a significant effect of MCD_lag (red bar in TOJ) may simply mean that the multisensory neural signal can be readily explained by MCD_V(t) minus MCD_A(t), and whether the non-significance of this effect in CJ task simply means that the multisensory neural signal in this task could instead only be better explained by a more flexible combination of MCD_V(t) and MCD_A(t) with different weights. The authors need to show that this is not the main account for the task difference in MCD_lag effect in Fig. 4.

In addition, I have a few minor suggestions on improving the clarity of the manuscript:

1) The MCD_corr and MCD_lag were averaged across time, but it is not clear how long the time window is. Is it the same as in Parise & Ernst (2016), i.e., 3 s?

2) Why is tau_A larger than tau_V across all participants in this study? This is inconsistent with the results in Parise & Ernst (2016): modality-dependent temporal constant of the filter tau_V = 87 ms and tau_A = 68 ms. This is particularly strange considering that the stimuli were generated using the parameter values in Parise & Ernst.

3) MCD_corr by its nature is non-negative, but is reported as a mean-subtracted or z-scored measure. This needs to be explained. It'd be better to have a Table in supplemental material reporting clearly the MCD_corr and MCD_lag in their original scale as closely to the ones used in Parise & Ernst (2016) as possible.

4) The description of the permutation tests seems unclear. Were the authors shuffling trial-wise task labels when doing permutation tests for the task contrast? Also, were maximum-statistics used to derive null distributions? I suggest that the authors follow (Nichols & Holmes, 2002) when reporting statistical tests.

5) The claim about unexpected RT effects needs to be toned down. The delay between the end of a stimulus and the "go" cue was about 1 s only. Considering that the duration of the stimulus was 1 s itself, this response delay is likely not long enough to completely dissociate action selection from sensory processes.

References:

Parise, Cesare V., and Marc O. Ernst. 2016. "Correlation Detection as a General Mechanism for Multisensory Integration." *Nature Communications* 7 (June): 11543.

Rohe, Tim, Ann-Christine Ehlis, and Uta Noppeney. 2019. "The Neural Dynamics of Hierarchical Bayesian Causal Inference in Multisensory Perception." *Nature Communications* 10 (1): 1907.

Nichols, Thomas E., and Andrew P. Holmes. 2002. "Nonparametric Permutation Tests for Functional Neuroimaging: A Primer with Examples." *Human Brain Mapping* 15 (1): 1–25.

Reviewer #2 (Remarks to the Author):

The authors of this work recorded MEG data while participants performed a temporal order judgment or a causality judgment task. They used the neural signal and applied the Multisensory Correlation Detector Model to see if there were some brain regions/signals that would account for these types of judgments. They found a region which was localized to the temporal-parietal cortices that fit well with the outputs of this model, particularly for the causality judgment task.

This manuscript is clear, concise, and these results are novel and have the potential to be quite impactful. Nevertheless, there are some issues noted below.

Major:

I am concerned about the use of ICA here to remove blinks. Given that the stimuli (at least half) are visual in nature, missing these stimuli because of blinking could both alter the judgment and the brain response. I suggest rejecting those epochs during which an eye blink occurred during the stimulus sequence.

Based on the behavioral data, and the TOJ task seems to have more stimuli that are perceived close to the 50/50 point than the causality judgment task. This is true for the group data and for the individual data, and the RTs were slower for the TOJ task as well. My concern here is that the ambiguity of these stimuli for this task is so great that the signals then being measured in the brain are uncertainty plus guess, rather than temporal detection per se. While I appreciate that the original model worked best on behavior close to this uncertainty, I am just wondering if the stimuli here could be resolved, or if there is just too much uncertainty.

While 13 participants may be adequate for a behavioral study, in MEG studies there are often more, and particularly given that the effect size isn't known a priori, it makes sense to be conservative and assume a smaller effect and run more participants. Please provide more of a justification for the sample size when it comes to the neural data.

On page 13 of the discussion, the authors state that they are comparing dynamics on a single-trial level and this is not related to the later decisional processes. I am not totally convinced that this is the case. Because the stimuli are not singular but rather sequences, the later activity could still be decision processes based on the initial part of the sequence. The authors need to provide a more convincing argument that they are really capturing the computation and not the decision as this is one of the main points that would make this work both novel and impactful.

Minor:

On page 17, please expand on what is meant in terms of correlated here in this paragraph for the stimuli themselves and for the task, and how exactly this was assessed.

In the equation on page 19, should there be a "V" instead of a "B"?

Please report the average number of trials per subject and the range after artifact rejection.

Please provide the exact timing of each sequence in the supplementary information.

Reviewer #1

Capitalizing upon an elegant computational model (Parise & Ernst, 2016), the authors of this manuscript sought to find neural correlates of “input-driven” model-based multisensory representations in the human brain. The behaviour of their human subjects can be well described by the multisensory correlation detector (MCD), replicating previous findings. Contrasting magnetoencephalographic data in different cognitive tasks on identical sensory inputs, the authors find task-related neural effects in and around parietal-temporal brain regions. A dynamic encoding model further reveals unisensory representations in respective unisensory cortices, neural representation of a correlation detector in parietal-temporal regions, and a temporal-order detector in frontal area. I think this manuscript makes a potential contribution to our understanding of how the physiologically-plausible MCD might actually be linked to the brain, thus extending that 2016 modelling paper. The dynamic encoding modelling is innovative and particularly suitable for addressing research questions about spatiotemporal dynamics in MEG data. However, I have several concerns pertaining to their analysis methods and data interpretation.

We thank Reviewer 1 for the general positive appreciation of the importance and novelty of our magneto-physiological work and encoding model methods.

Major:

1) The MCD model parameters, i.e., 3 time constants, were estimated by fitting the model to both tasks at the same time. I find this decision unwarranted. Why are the time constants a priori assumed to be unchanged across tasks? Fig. 2 shows that the psychometric functions differ between the two tasks. I worry a changing time constant(s) across the tasks would lead to similar results. Here is a suggestion on how to justify your choice of the fixed model parameters across tasks: instead of fitting the model to your data by maximising the correlation between empirical vs. model-predicted psychophysical kernel, which relies on collapsing trials, you can fit the model to trial-by-trial behavioural responses by maximising binomial log likelihood. You can include a noise parameter for the transfer function linking MCD to choice e.g., a softmax or cumulative Gaussian, on a trial-by-trial basis. You can fit this model (4 free parameters: 3 time constants + 1 noise parameter) to each task independently, and see if any parameter differs systematically across tasks. It's been well-known that, to learn and make decisions appropriately, humans and other animals have evolved to integrate various sensory information on multiple timescales. These time constants are essentially geared to capture such flexible timescales in the model and are likely to change as a function of cognitive tasks. This concern greatly affects the interpretation of their neural results.

This point is well taken, and when the MCD model was first developed, the authors wondered whether task-related effects may alter such estimated parameters. To test whether this was the case, two experiments were run in the original study (Parise & Ernst, 2016). In the main experiment, participants were required to perform a dual task and jointly provide temporal order and causality judgments on each trial. The parameters of the model were fitted with the data of such a dual task experiment. Next, the authors performed a control experiment on a new set of participants using a single-task block-design, whereby on each

block participants only provided temporal order and causality judgments (depending on the block). Using the parameters fitted in the dual task experiment, the authors managed to tightly predict the performance on the single task experiment: that is, the correlation between predicted and actual results was 0.92 and 0.98 for the causality and temporal order judgments, respectively (see Parise & Ernst, 2016, Supplementary Figure 9). Based on such evidence, it seems reasonable to conclude that the experimental task only has negligible (if any) effects on the activity of the proposed multisensory correlation detectors, hence making it appropriate in the current study to jointly fit the parameters on both tasks, as that allows to perform the fitting on a much larger sample size.

2) Motor-activity/response-bias confounds. Across the two tasks, the response-choice mapping was fixed in all blocks for all subjects. In causality judgment, subjects used the right index finger for “same cause” choice and the right middle finger for “different cause”; in temporal-order judgment, they used the right index finger for “sound leading” choice and the right middle finger for “vision leading”. This design would lead to two potential confounds when interpreting the MEG results in Fig. 3. **Firstly**, the neural effect of task change may reflect change in motor effect or response bias. It is very possible that, for the same stimulus, a subject used the index finger to commit a “same cause” choice in one task and the middle finger for a “vision leading” choice in the other task. Simple solutions such as 1) including binary finger code as covariates in the task contrast analysis or 2) running the task contrast only for trials with identical finger response would rule out this confound. But the data in Fig. 3A so far can’t be interpreted as a genuine neural effect of task change, above and beyond motor-related signal change. Of note, those significant ‘white sensors’ in Fig. 3A (also in Supp Fig. 4A) are mostly in the left hemisphere, which may be echoing this motor confound. **Secondly**, the effect of model-predicted MCD_corr on MEG activity (Fig. 3B left) may reflect motor effect because MCD_corr is a strong predictor for choice (= finger). The striking similarity between the wavy pattern in Fig. 3B left and that in Fig. 2B indicates that the MEG activities may indeed reflect finger identity.

Following Reviewer 1’s concerns, we have decided to decompose our task contrast analysis (2: causality vs. temporal order judgment) finger by finger (2: middle vs. index finger) thus sorting four sets of evoked responses. We then ran a spatiotemporal clustering analysis using ANOVA (Task identity x Finger identity). This supplementary analysis did not reveal a significant effect of Finger identity ($p > 0.05$ for all clusters on the F-values associated with Finger identity) nor a significant effect of the interaction ($p > 0.05$ for all clusters on the F-values associated with the interaction Task identity x Finger identity).

As this may be of concern for several readers, we have added this complementary analysis in the Results and added a Supplementary Figure:

“This result was robust to two possible biases: [...] and second, the lack of randomization of the response choice (fixed across blocks and participants) may contribute to the effect. We contend that this non-randomization could have introduced a motor response and/or decisional bias in our results. We demonstrate the persistence of this significant cluster of activity [...] when decomposing the task contrast analysis by response choice (see Supp. Fig. 6).”

“**Supplementary Figure 6.** Spatiotemporal-cluster analysis contrasting brain activity evoked by the presentation of identical audiovisual sequences but in two different tasks (causality vs. temporal order judgment). Left: t-map of the significant cluster (white sensors, $p < 0.05$ corrected for multiple comparison) ranging from 260 to 1250 ms post-sequence onset. Grey levels are t-values averaged across significant times. Right: Temporal extent of the effect averaged over significant sensors in the cluster (grey). The top and bottom panels represent the two polarities of a single source (positive left, negative right). This analysis replicates the results presented in Fig. 3A. A spatiotemporal clustering analysis on a 2 by 2 ANOVA with factor Task identity (2) and Finger identity (2) showed no significant effects of Finger identity ($p > 0.05$ for all clusters on the F-values associated with Finger identity) or interaction between Task and Finger identity ($p > 0.05$ for all clusters on the F-values associated with the interaction Task identity x Finger identity).”

Additionally, we realized that a small mistake on our part in Fig. 3A and Supp. Fig. 4A may have accidentally led Reviewer 1 towards an incorrect conclusion concerning a possible left lateralization of the effect. Indeed, only the left part of the cluster was shown in both figures whereas the clusters are bilateral. We have corrected this mistake in both Fig. 3A:

(previous Figure 3A)

Corrected Figure 3A.

... and Supp. Fig. 4A:

(previous Supplementary Figure 4A)

Corrected Supplementary Figure 4A.

Finally, we believe that the replication of the task contrast analysis by the model-based analysis rules out the possibility of a motor confound. Indeed, the model-based analysis is not based on a simple contrast (causality vs. temporal order judgment) but on the

similarity across the whole epoch between the predicted signals and the signals recorded through MEG. A motor confound would manifest as a global amplitude or latency difference between conditions, not as a signal that happens to resemble the MCD model.

3) Another concern is about regressing out unisensory signals from multisensory signals. Of note, the authors did this regression separately for each task. This means that the residuals in the multisensory signal are those residual activities that cannot be explained by any weighted combination of the unisensory auditory and visual signals alone. However, it has been known that change of cross-modal decision strategies can be reflected by the change of linear neural weight assigned to each modality (Rohe, Ehrlis & Noppeney, 2019 Nat. Comm.). If one regresses out the flexible combination of unisensory signals from each task separately by allowing 4 free weights (2 for A/V in causality task and 2 in temporal order judgment task), the residuals in the multisensory condition would completely miss out the chance of describing any neural effect in the brain that potentially corresponds to this task-related weight change. Even though I understand that one may reasonably assume that only the 'nonlinear' effect (above and beyond weighted average) is interesting when speaking of multisensory interaction, I find this residualization approach unwarranted. For instance, the authors made a strong assumption that the brain signals evoked by unisensory inputs can be readily generalisable to multisensory conditions as 'component' signals at the exact MEG sensor and at the exact moment in time (this is how the residualization was done). But it is unclear if the spatiotemporal pattern may have been altered by change of attention etc. across unisensory vs. multisensory conditions, which makes that assumption a bit shaky.

We understand that Reviewer 1 is concerned by the fact that removing the weighted combination of the unisensory auditory and visual signals might have introduced some bias and/or removed some relevant signal.

To address this issue, we ran a supplementary analysis in which we do not perform the linear regression step and we do not remove the unisensory signals from the multisensory recordings before computing the spatiotemporal clustering analysis. Doing so, the results are very similar to what we report in the original manuscript (see Supp. Fig. 5): we found two significant clusters (each polarity of the same source) peaking around 700 ms following the onset of the audiovisual sequence: [200 - 1250 ms] [375 - 1200 ms] in the left and right sensors, respectively.

We have added this complementary analysis in the Results and added a Supplementary Figure:

"This result was robust to two possible biases: first, the unisensory-free data may be biased by the weighting process [...]. We demonstrate the persistence of this significant cluster of activity without removing the unisensory-specific activity (see Supp. Fig. 5)."

“**Supplementary Figure 5.** Spatiotemporal-cluster analysis contrasting brain activity evoked by the presentation of identical audiovisual sequences but in two different tasks (causality vs. temporal order judgment). Left: t-map of the significant cluster (white sensors, $p < 0.05$ corrected for multiple comparison) ranging from 200 to 1250 ms post-sequence onset. Grey levels are t-values averaged across significant times. Right: Temporal extent of the effect averaged over significant sensors in the cluster (grey). The top and bottom panels represent the two polarities of a single source (positive left, negative right). This analysis replicates the results presented in Fig. 3A, using the signals without performing the removal of the unisensory-specific activity.”

4) $MCD_{corr}(t)$ is the product of $MCD_A(t)$ and $MCD_V(t)$, whereas $MCD_{lag}(t)$ is the linear combination of them. This in fact would encourage alternative interpretation of the results in Fig. 4. It is unclear whether a significant effect of MCD_{lag} (red bar in temporal order judgment) may simply mean that the multisensory neural signal can be readily explained by $MCD_V(t)$ minus $MCD_A(t)$, and whether the non-significance of this effect in CJ task simply means that the multisensory neural signal in this task could instead only be better explained by a more flexible combination of $MCD_V(t)$ and $MCD_A(t)$ with different weights. The authors need to show that this is not the main account for the task difference in MCD_{lag} effect in Fig. 4.

Reviewer 1 is correct in stating that, if MCD_{LAG} is a linear combination of MCD_A and MCD_V , then MCD_{LAG} does not add anything to the linear model used in Fig. 4. If so, this would explain why MCD_{LAG} does not have a significant effect, especially in the CJ task. However, MCD_{LAG} is not a linear combination of MCD_A and MCD_V . Formally:

$$MCD_{LAG} = (MCD_A \otimes f_{AV}) \cdot MCD_V - (MCD_V \otimes f_{AV}) \cdot MCD_A$$

(where \otimes is the convolution operator and f_{AV} is an exponential decay filter)

We believe that this slight misunderstanding comes from a bad choice of symbols in the schema of the model in Fig. 1 and Fig. 4. To prevent any further misinterpretation, we have harmonized the representation of the nodes so that it is clearer that the signals are filtered multiple times before being multiplied (MCD_{CORR}) or subtracted (MCD_{LAG}). What is being multiplied and subtracted is not MCD_A or MCD_V directly but rather a filtered and mixed version of both. We have corrected Fig. 1 and Fig. 4 accordingly.

Multisensory Correlation Detector (MCD)

Figure 1. The Multisensory Correlation Detector is a set of low-pass filters (filled grey boxes), which compute the temporal correlation (MCD_{CORR} , blue) and the temporal lag (MCD_{LAG} , red) between incoming multisensory signals. The Multisensory Correlation Detector model is biologically plausible as it integrates signals based on their cross-correlation in time and implements Bayesian optimal cue combination. LPF: low-pass filter.

Furthermore, we have added the complete mathematical description of the Multisensory Correlation Detector:

“Formally, the Multisensory Correlation Detector is composed of a first filtering stage, where time-varying visual and auditory signals (S_A , S_V) are independently low-pass filtered, and a subsequent integration stage, where the two signals are combined through linear operations (multiplication or subtraction). Low-pass filters were modelled as exponential functions with τ_{mod} as the modality-dependent time constant:

$$f_{mod}(t) = t \cdot e^{\frac{-t}{\tau_{mod}}}$$

$$MCD_A(t) = f_A(t) \otimes S_A(t)$$

$$MCD_V(t) = f_V(t) \otimes S_V(t)$$

Each subunit (MCD_{S_1} , MCD_{S_2}) of the detector independently combines multisensory information by multiplying the filtered visual and auditory signals as follows:

$$MCD_{S_1}(t) = (MCD_A(t) \otimes f_{AV}(t)) \cdot MCD_V(t)$$

$$MCD_{S_2}(t) = (MCD_V(t) \otimes f_{AV}(t)) \cdot MCD_A(t)$$

The response of the subunits are eventually multiplied or subtracted:

$$MCD_{CORR}(t) = MCD_{S_1}(t) \cdot MCD_{S_2}(t)$$

$$MCD_{LAG}(t) = -MCD_{S_1}(t) + MCD_{S_2}(t)”$$

In addition, I have a few minor suggestions on improving the clarity of the manuscript:

1) The MCD_corr and MCD_lag were averaged across time, but it is not clear how long the time window is. Is it the same as in Parise & Ernst (2016), i.e., 3 s?

We confirm that the time window was the same as in Parise & Ernst (2016). We have added this detail to the Methods as follows:

“The MCD_{CORR} and MCD_{LAG} values were time-averaged in a 3s-long time window — three times the duration of each sequence —”.

2) Why is tau_A larger than tau_V across all participants in this study? This is inconsistent with the results in Parise & Ernst (2016): modality-dependent temporal constant of the filter tau_V = 87 ms and tau_A = 68 ms. This is particularly strange considering that the stimuli were generated using the parameter values in Parise & Ernst.

We thank Reviewer 1 for spotting this typo. The labels of the table were inverted. We confirm that τ_V is larger than τ_A for all participants and that the values are on average close to those reported in Parise & Ernst (2016), with $\tau_{AV} = 746$ ms, $\tau_V = 126$ ms and $\tau_A = 69$ ms. We also confirm that this has not impacted our results and conclusions because the values used for the analysis were correct (it is only the table in the manuscript that was wrongly reported). We have corrected this mistake:

ID	1	2	3	4	5	6	7	8	9	10	11	12	13
τ_V (ms)	130	137	120	152	142	141	122	123	114	109	103	119	122
τ_A (ms)	84	80	58	110	92	73	50	53	52	52	54	64	74
τ_{AV} (ms)	519	490	562	1147	882	1087	748	372	764	779	799	779	770

Table 1. Fitted parameters of the Multisensory Correlation Detector for each participant.

3) MCD_corr by its nature is non-negative, but is reported as a mean-subtracted or z-scored measure. This needs to be explained. It'd be better to have a Table in supplemental material reporting clearly the MCD_corr and MCD_lag in their original scale as closely to the ones used in Parise & Ernst (2016) as possible.

We have chosen to z-score the values to allow an easier comparison between the effect and MCD_{CORR} and MCD_{LAG} on behavior and on the MEG signals. Further, it improves the stability of the linear regression and reduces the correlation between predictors. The unnormalized values can now also be found in Supp. Table 1:

“These values were then z-scored across stimuli to allow comparing more easily the effects of each on behavior and on MEG activity. The unnormalized values can be found in Supp. Table 2.”

Sequence #	MCD _{CORR}	MCD _{LAG}
1	15.9	-0.41
2	15.0	0.37
3	15.4	0.96
4	14.7	0.42
5	15.4	0.20
6	16.2	0.36

Supplementary Table 2. Unnormalized values of MCD_{CORR} and MCD_{LAG} of the sequences used in the experiment.

4) The description of the permutation tests seems unclear. Were the authors shuffling trial-wise task labels when doing permutation tests for the task contrast? Also, were maximum-statistics used to derive null distributions? I suggest that the authors follow (Nichols & Holmes, 2002) when reporting statistical tests.

We now follow the recommendations of Nichols & Holmes (2002) to report the spectro-temporal permutation procedure. Note that the method is very classical and endorsed by the community for the analysis of MEG contrasts (Maris & Oostenveld, 2007, cited > 5000 times; Gross et al (2013). Good practice for conducting and reporting MEG research. *Neuroimage*, 65, 349-363.).

“The significance of the contrasts was assessed using non-parametric pairwise two-tailed permutation tests across time and sensors, which provided corrected p-values for multiple comparisons. The null hypothesis is that the MEG amplitude should be the same whatever the task. The statistic of interest in the t-value of the paired t-test contrasting the task across participants and stimuli. Spatio-temporal clusters were defined on the basis of temporal adjacency and covariance matrix by regrouping samples whose T-statistic was larger than 2.6, corresponding to a p-value ≤ 0.01 for a T-test with 77 degrees of freedom (13 participants x 6 sequences – 1). Cluster-level statistics were then calculated by taking the sum of the t-values within the cluster. We used random permutations to build the null distribution of the cluster-level statistics. On each permutation (N = 10 000), task labels were randomly shuffled across trials (Maris & Oostenveld, 2007). Only spatio-temporal clusters with permutation corrected p-values ≤ 0.01 are reported.”

5) The claim about unexpected RT effects needs to be toned down. The delay between the end of a stimulus and the “go” cue was about 1 s only. Considering that the duration of the stimulus was 1 s itself, this response delay is likely not long enough to completely dissociate action selection from sensory processes.

We agree with Reviewer 1 that the RT should be interpreted with caution, especially because the experiment was not built to interpret those. We have now added a note of caution in the Supp. Fig. 2, where this side result is presented:

“Nonetheless, considering that the response delay was 1s, it is likely not long enough to completely dissociate action selection from sensory processes.”

References:

Parise, Cesare V., and Marc O. Ernst. 2016. "Correlation Detection as a General Mechanism for Multisensory Integration." *Nature Communications* 7 (June): 11543.

Rohe, Tim, Ann-Christine Ehlis, and Uta Noppeney. 2019. "The Neural Dynamics of Hierarchical Bayesian Causal Inference in Multisensory Perception." *Nature Communications* 10 (1): 1907.

Nichols, Thomas E., and Andrew P. Holmes. 2002. "Nonparametric Permutation Tests for Functional Neuroimaging: A Primer with Examples." *Human Brain Mapping* 15 (1): 1–25.

Reviewer #2

The authors of this work recorded MEG data while participants performed a temporal order judgment or a causality judgment task. They used the neural signal and applied the Multisensory Correlation Detector Model to see if there were some brain regions/signals that would account for these types of judgments. They found a region which was localized to the temporal-parietal cortices that fit well with the outputs of this model, particularly for the causality judgment task.

This manuscript is clear, concise, and these results are novel and have the potential to be quite impactful. Nevertheless, there are some issues noted below.

We thank Reviewer 2 for the positive appreciation on the novelty and potential impactfulness of the study.

Major:

1) I am concerned about the use of ICA here to remove blinks. Given that the stimuli (at least half) are visual in nature, missing these stimuli because of blinking could both alter the judgment and the brain response. I suggest rejecting those epochs during which an eye blink occurred during the stimulus sequence.

The Reviewer fears that participants are missing some of the visual stimuli because of blinking. We understand this concern. However we would like to gently disagree with Reviewer 2 on this matter. First, participants were explicitly told to refrain from blinking during the presentation of the audiovisual sequences, as stated in the Methods:

"[Participants] were asked to refrain from blinking during the presentation of the audiovisual sequences."

As a result, participants blinked on average 0.1 times per sequence, which equates to missing 1 flash every 50 on average. We believe that this is clearly not sufficient to critically impair the judgment and brain responses of the participants.

Second, large artifacts, corresponding to head movement or large eye blinks, were automatically removed using peak-to-peak amplitude rejection:

"Epochs contaminated by artifacts were rejected based on the peak-to-peak amplitude ($2 \cdot 10^{-11}$ T for magnetometers and $9 \cdot 10^{-9}$ T.cm⁻² for gradiometers)"

Third, if participants were blinking too much and missing a lot of visual stimuli, this would manifest as a Type II error (missing a true effect). However, we do find that the audiovisual sequences have the expected effect on the behavior and on the MEG activity of the participants.

Finally, concerning the method of artifacts removal, note that ICA is the gold standard to remove blinks artifacts (Gross et al., *Neuroimage*, 2013) and that our method using

mne-python tools carefully tailors the ICA based on an individualized template method thereby minimizing the possibility to remove signal with noise:

“Data preprocessing was done in accordance with accepted guidelines for MEG research (Gross et al., 2013).

2) Based on the behavioral data, and the TOJ task seems to have more stimuli that are perceived close to the 50/50 point than the causality judgment task. This is true for the group data and for the individual data, and the RTs were slower for the TOJ task as well. My concern here is that the ambiguity of these stimuli for this task is so great that the signals then being measured in the brain are uncertainty plus guess, rather than temporal detection per se. While I appreciate that the original model worked best on behavior close to this uncertainty, I am just wondering if the stimuli here could be resolved, or if there is just too much uncertainty.

We thank Reviewer 2 for this insightful comment. Following Reviewer 2 's comment, we have built a reduced dataset containing only the stimuli that are ambiguous in the temporal order judgment task (stimuli blue, green, orange and red in Fig. 2). We then replicated the model-based analysis presented in Fig. 4 in this reduced dataset. The results show that the model is performing equally well in this reduced dataset. We have added this result in the manuscript:

“One further limitation could be that 4 out of 6 stimuli are perceived close to the 50/50 point in the temporal order judgment blocks. It could be the case that the ambiguity of these stimuli for this task is so great that the signals being measured in the brain are uncertainty plus guess, rather than temporal order computations per se. To rule out this hypothesis, we replicated the analysis in a reduced dataset comprising only the stimuli that are ambiguous in the temporal order judgment task (see Fig. Supp. 8). The results show that the temporal response functions remain stable in this reduced dataset and that the results remain unchanged compared to the complete dataset. This rules out the possibility that the signal being recorded reflects only uncertainty or guesses.”

Supplementary Figure 8. Model-based temporal response functions, reduced dataset. A. Replication of the data presented in Fig 4 for the 4 stimuli that receive uncertain judgment in the temporal order judgment task (stimuli blue, green, orange and red in Fig. 2). Statistical significance of the correlation between cross-validated predicted MEG and true MEG was assessed via corrected cluster permutations. Significant sensors are highlighted in white. Error bars represent 2 s.e.m. across participants. Dots represent individual participants.

Nonetheless, we agree with the Reviewer that the high ambiguity of the stimuli on the temporal order judgment could have contributed to the absence of detection of an effect of MCD_{LAG} in the temporal order judgment blocks. We have added this limitation in the interpretation of our results in the Discussion:

“Furthermore, 4 out of 6 stimuli were perceived as ambiguous with regard to the temporal order judgment. Such a high level of uncertainty could have reduced the detectability of the MCD_{LAG} effect on the MEG recordings.”

3) While 13 participants may be adequate for a behavioral study, in MEG studies there are often more, and particularly given that the effect size isn't known apriori, it makes sense to be conservative and assume a smaller effect and run more participants. Please provide more of a justification for the sample size when it comes to the neural data.

We understand that the Reviewer is concerned by the seemingly small number of participants tested here. However, considering the large number of trials run by each participant, a pool of 13 participants is indeed both more than sufficient to support all the claims in this manuscript and in line with similar model-based MEG studies on multisensory integration (see for example, Cao et al., *Neuron*, 2019). To properly support this claim here we present: (1) a sensitivity analysis to demonstrate how our conclusions would be affected by both the total number of trials and participants, (2) single-subject analyses, showing the consistency of our results across participants, and (3) a review of the current literature on model based neuroimaging studies on multisensory integration.

(1) We ran the spatiotemporal clustering analysis (the main analysis of the paper, presented in Fig. 3) with an artificially reduced dataset. We randomly removed $N_s = 1, 2, 3, \dots$ participants and $N_t = 30, 45, 60 \dots$ repetitions of each sequence and computed the t -statistics in the spatiotemporal cluster that we have previously identified. We performed this procedure 100 times for each combination of N_s removed participants and N_t removed trials. The averaged values across the 100 times are displayed in the newly added Fig. Supp. 9.

Supplementary Figure 9. Average t -value inside the spatiotemporal cluster for randomly selected $N_s = 4, 5, \dots, 13$ participants and $N_t = 30, 45, \dots, 150$ repetitions of each sequence. The procedure was repeated 100 times and averaged across the 100 repetitions. Any combination of number of participants and number of repetitions per participant above the red line (which corresponds to a total of 10,500 trials) would be sufficient to show an effect (we have 19,500 trials). The red dot indicates the current study. Dots (black and red) indicate a significant t -value ($p < 0.05$).

Based on these simulations, we can see that even 7 participants with 150 repetitions of each sequence would have been sufficient to detect the effect. Similarly, only 30 repetitions of each sequence with 13 participants would have been sufficient to detect the effect. More generally, any combination of number of participants and number of repetitions per participant above the diagonal red line would be sufficient to support our claims. This corresponds to a total of 10,500 trials. We have a total of 19,500 trials, which is about twice as much. We are therefore confident that 13 participants with 150 repetitions of each sequence is an adequate sample size for our study.

(2) Second, we assessed how stable the reported effects were in single participants and added this information in a novel Fig. 4 to show the individual values for the model-based results. The effects we report are present in 13 out of 13 participants (except MCD_{LAG} in temporal order judgment blocks that concerns 10 out of 13 participants).

Extract of Figure 4. Stimulus- vs. Model-based temporal response functions. [...] Overall, MCD_A, MCD_V and MCD_{CORR} explain MEG variance consistently across causality and temporal order judgment blocks. To the contrary, MCD_{LAG} significantly fits the data only in the temporal order judgment blocks. Statistical significance of the correlation between cross-validated predicted MEG and true MEG was assessed via corrected cluster permutations. Error bars represent 2 s.e.m. across participants. Dots represent individual participants. [...]

(3) Finally, our total number of trials (19,500) is in the same range as what has been recently reported. For example, a recent paper using similar methods – model-based analyses on MEG data to study multisensory processing – reports 16 participants and a total of 26,752 trials (Cao et al., *Neuron*, 2019). Similarly, there are several papers using fMRI to study multisensory processing that report less than half of our participants: 6 participants in Rohe et al. (*Current Biology*, 2016) and 6 participants in Rohe et al. (*PLoS Biology*, 2015). Generally speaking, we preferred to have fewer participants but a larger number of trials per participant as in MEG (just like fMRI) is very sensitive to head displacement between participants. All in all, we strongly believe that such a comparison with the state of the art, along with the analyses reported above both justify and empirically support the adequacy of our total sample size. We have added these justification for our sample size in the Methods:

“The sample size was determined on the basis of the original paper describing the psychophysical task and main effects (Parise & Ernst, 2016, N = 5) and is comparable to studies using similar stimuli and paradigms (for example Cao et al., 2019).”

4) On page 13 of the discussion, the authors state that they are comparing dynamics on a single-trial level and this is not related to the later decisional processes. I am not totally convinced that this is the case. Because the stimuli are not singular but rather sequences, the later activity could still be decision processes based on the initial part of the sequence. The authors need to provide a more convincing argument that they are really capturing the computation and not the decision as this is one of the main points that would make this work both novel and impactful.

Reviewer 2 is correct that it is important to show that the MEG effects are not related to decisional processes but to the computations themselves. To do so, we split the evoked response in two halves: an early part, from 0 to 500 ms, and a late part, from 500 to 1000 ms. We replicated the model-based analysis (presented in Fig. 4) on these two halves. In the first half, we found the same results as with the full evoked responses, plus a significant effect of MCD_{LAG} in the temporal order judgment blocks. In the second half, we found the same results as with the full evoked responses, minus a significant effect of MCD_A in CJ blocks. Overall, the results show that the TRFs are stable across the two halves and consistent with what we report for the full evoked responses. This rules out the possibility that this activity reflects decisional processes, as decisional processes are supposed to

happen only late in the sequence presentation. We have added a paragraph in the Results and a Supplementary Figure:

“However, as the stimuli are not singular but sequences, later activity could at least partly be decisional processes based on the initial part of the sequence and not entirely related to the computations themselves. To rule out this hypothesis, we replicated the analysis by splitting the evoked response in two halves: an early part, from 0 to 500 ms, and a late part, from 500 to 1000 ms (see Fig. Supp. 7). The results show that the temporal response functions are stable across the two halves. This rules out the possibility that this activity reflects decisional processes, as decisional processes are supposed to happen only late in the sequence presentation.”

A Explained variance per model units, blocks and MEG sensors (0 to 500 ms)

B Explained variance per model units, blocks and MEG sensors (500 to 1000 ms)

Supplementary Figure 7. Model-based temporal response functions, split analysis. **A.** Replication of the data presented in Fig 4 for the first half of the evoked response (from 0 to 500 ms). **B.** Replication of the data presented in Fig 4 for the second half of the evoked response (from 500 to 1000 ms). Statistical significance of the correlation between cross-validated predicted MEG and true MEG was assessed via corrected cluster permutations. Significant sensors are highlighted in white. Error bars represent 2 s.e.m. across participants. Dots represent individual participants.

Minor:

1) On page 17, please expand on what is meant in terms of correlated here in this paragraph for the stimuli themselves and for the task, and how exactly this was assessed.

We now precise what is meant in terms of correlation on page 17:

“Indeed, in order to study the unique contribution of neural sources to causality and temporal order judgments, it is critical to (1) use the same stimuli and (2) to have uncorrelated judgments across stimuli (i.e., stimuli associated with a given causality judgment should be systematically associated with a given temporal order judgment)”.

“We then selected six sequences among these 10^6 to elicit a large range of responses of MCD_{CORR} and MCD_{LAG} values, while maintaining the correlation between MCD_{CORR} and MCD_{LAG} low (Pearson’s correlation between MCD_{CORR} and MCD_{LAG} across the six sequences below 0.2).”

2) In the equation on page 19, should there be a “V” instead of a “B”?

We thank the Reviewer for spotting this small mistake. We have corrected it.

3) Please report the average number of trials per subject and the range after artifact rejection.

We have added this precision in the Methods:

“Across participants, an average of 98.9% of epochs were kept in the analyses.”

4) Please provide the exact timing of each sequence in the supplementary information.

We have now added this information in a new Supp. Table 1:

Sequence #	1 st sound/flash (ms)	2 nd	3 rd	4 th	5 th
1	53	194	372	511	800
	288	494	601	731	834
2	118	214	693	752	807
	163	364	524	614	940
3	338	397	534	734	844
	104	188	262	442	807
4	65	184	238	646	883
	117	184	322	413	469
5	108	298	489	739	900
	128	278	505	783	879
6	64	165	491	587	645
	58	130	363	624	788

Supplementary Table 1. Precise timing (ms) of the sequences used in the experiment.

REVIEWERS' COMMENTS

Reviewer #1 (Remarks to the Author):

The authors have done a good job addressing the technical concerns that I outlined. Following my suggestion, their new control analysis for ruling out motor confounds is very meaningful. I like Supplementary Figure 6 and I am glad the task effect is not confused by the finger identity. But there is a typo in the legend of this figure: The dark-orange curve should be TOJ_middle instead? Otherwise both the dark- and light- orange curves are denoted as TOJ_index, which makes no sense.

I have no other technical concerns.

I liked this study, and I think it has the potential to make a contribution to the literature. But I was surprised that in the Discussion this study didn't discuss much about the discrepancy between their neural effects and the previous literature on multisensory causal inference. There has been an ongoing debate over whether multisensory causal inference recruits frontal regions or just anterior parietal cortex (Mihalik & Noppeney, 21; Fang et al., 19; Cao et al., 19; Rohe & Noppeney, 16). Interestingly, the current study has found both frontal and parietal effects, but the frontal effect is weaker and is not found in the explicit TOJ task. But currently this paper doesn't discuss this sufficiently. The frontal cortex, particularly the lateral PFC has been widely found as a key region for multisensory convergence and multisensory conflict resolution (Noppeney, Ostwald, & Werner, 10; Romanski, 12). I think the discrepancy could be in part due to different analytical approaches. Most previous studies (that have revealed strong frontal effects) have relied on multivariate encoding/decoding analyses, whereas the "temporal response function analysis" in the current study is a univariate approach (e.g., it models the activity at each MEG channel individually). Of note, Mihalik & Noppeney (2021) could find the lateral PFC as the only region encoding causal decision in an explicit causal inference task (very similar to the one used here) but only when relying on a multivariate decoding approach. In mass-univariate analysis on the same data, they could only find the parietal effects. In their univariate results, the frontal cortex doesn't respond to the causal reports per se, but the frontal cortex does respond to an interaction between causal report and stimulus' physical congruency. The analysis in this study is in the spirit of a univariate approach to assessing neural effects of MCD_corr (highly relevant for causal decision) in the TOJ task.

Overall, I think this paper has the potential to make a more important impact if the Discussion could include some of these points or beyond, so that it doesn't look like just a 'brain' story of the Multisensory Correlation Detector (obviously the only computational model the authors considered). I know that when authors include a computational model in their paper, they inevitably brace themselves for the reviewers coming back with the suggestion that they additionally test multiple alternative models that they may not have considered. This is sometimes onerous and often unnecessary for the

integrity of the paper. I am not going to do that here. But, I do hope to see a little bit more insightful discussions on the neural effects reported here and their link to the literature of the neural basis of multisensory causal inference.

References:

Fang, W., Li, J., Qi, G., Li, S., Sigman, M., & Wang, L. (2019). Statistical inference of body representation in the macaque brain. *Proceedings of the National Academy of Sciences*, 116(40), 20151-20157.

Mihalik, A., & Noppeney, U. (2020). Causal inference in audiovisual perception. *Journal of Neuroscience*, 40(34), 6600-6612.

Cao, Y., Summerfield, C., Park, H., Giordano, B. L., & Kayser, C. (2019). Causal inference in the multisensory brain. *Neuron*, 102(5), 1076-1087.

Rohe, T., & Noppeney, U. (2016). Distinct computational principles govern multisensory integration in primary sensory and association cortices. *Current Biology*, 26(4), 509-514.

Noppeney, U., Ostwald, D., & Werner, S. (2010). Perceptual decisions formed by accumulation of audiovisual evidence in prefrontal cortex. *Journal of Neuroscience*, 30(21), 7434-7446.

Romanski, L. M. (2012). Convergence of auditory, visual, and somatosensory information in ventral prefrontal cortex. In: *The neural bases of multisensory processes* (Murray M, Wallace M, eds), pp 667–682. Boca Raton (FL): CRC Press.

Reviewer #2 (Remarks to the Author):

The authors have addressed my concerns in their revision, and I appreciate their thorough responses.

Reviewer #1

The authors have done a good job addressing the technical concerns that I outlined. Following my suggestion, their new control analysis for ruling out motor confounds is very meaningful. I like Supplementary Figure 6 and I am glad the task effect is not confused by the finger identity. But there is a typo in the legend of this figure: The dark-orange curve should be TOJ_middle instead? Otherwise both the dark- and light- orange curves are denoted as TOJ_index, which makes no sense.

We thank Reviewer 1 for his/her comments. We have corrected the typo in the legend of Supplementary Figure 6.

I have no other technical concerns.

I liked this study, and I think it has the potential to make a contribution to the literature. But I was surprised that in the Discussion this study didn't discuss much about the discrepancy between their neural effects and the previous literature on multisensory causal inference. There has been an ongoing debate over whether multisensory causal inference recruits frontal regions or just anterior parietal cortex (Mihalik & Noppeney, 21; Fang et al., 19; Cao et al., 19; Rohe & Noppeney, 16). Interestingly, the current study has found both frontal and parietal effects, but the frontal effect is weaker and is not found in the explicit TOJ task. But currently this paper doesn't discuss this sufficiently. The frontal cortex, particularly the lateral PFC has been widely found as a key region for multisensory convergence and multisensory conflict resolution (Noppeney, Ostwald, & Werner, 10; Romanski, 12). I think the discrepancy could be in part due to different analytical approaches. Most previous studies (that have revealed strong frontal effects) have relied on multivariate encoding/decoding analyses, whereas the "temporal response function analysis" in the current study is a univariate approach (e.g., it models the activity at each MEG channel individually). Of note, Mihalik & Noppeney (2021) could find the lateral PFC as the only region encoding causal decision in an explicit causal inference task (very similar to the one used here) but only when relying on a multivariate decoding approach. In mass-univariate analysis on the same data, they could only find the parietal effects. In their univariate results, the frontal cortex doesn't respond to the causal reports per se, but the frontal cortex does respond to an interaction between causal report and stimulus' physical congruency. The analysis in this study is in the spirit of a univariate approach to assessing neural effects of MCD_corr (highly relevant for causal decision) in the TOJ task.

Overall, I think this paper has the potential to make a more important impact if the Discussion could include some of these points or beyond, so that it doesn't look like just a 'brain' story of the Multisensory Correlation Detector (obviously the only computational model the authors considered). I know that when authors include a computational model in their paper, they inevitably brace themselves for the reviewers coming back with the suggestion that they additionally test multiple alternative models that they may not have considered. This is sometimes onerous and often unnecessary for the integrity of the paper. I am not going to do that here. But, I do hope to see a little bit more insightful discussions on the neural effects reported here and their link to the literature of the neural basis of multisensory causal inference.

References:

Fang, W., Li, J., Qi, G., Li, S., Sigman, M., & Wang, L. (2019). Statistical inference of body representation in the macaque brain. *Proceedings of the National Academy of Sciences*, 116(40), 20151-20157.

Mihalik, A., & Noppeney, U. (2020). Causal inference in audiovisual perception. *Journal of Neuroscience*, 40(34), 6600-6612.

Cao, Y., Summerfield, C., Park, H., Giordano, B. L., & Kayser, C. (2019). Causal inference in the multisensory brain. *Neuron*, 102(5), 1076-1087.

Rohe, T., & Noppeney, U. (2016). Distinct computational principles govern multisensory integration in primary sensory and association cortices. *Current Biology*, 26(4), 509-514.

Noppeney, U., Ostwald, D., & Werner, S. (2010). Perceptual decisions formed by accumulation of audiovisual evidence in prefrontal cortex. *Journal of Neuroscience*, 30(21), 7434-7446.

Romanski, L. M. (2012). Convergence of auditory, visual, and somatosensory information in ventral prefrontal cortex. In: *The neural bases of multisensory processes* (Murray M, Wallace M, eds), pp 667–682. Boca Raton (FL): CRC Press.

We have added a paragraph in the Discussion concerning the neural basis of multisensory causal inference:

“A recent debate has taken place over whether multisensory causal inference recruits frontal regions or solely anterior parietal cortex (Mihalik and Noppeney 2020; Rohe and Noppeney 2016; Cao et al. 2019; Chandrasekaran 2017). The lateral prefrontal cortex has been reported multiple times as a key region for multisensory convergence and multisensory conflict resolution (Noppeney et al. 2010; Romanski 2012; Cléry et al. 2017; Coen et al. 2021). It is also a region suggested to play a key role in reverse hierarchical learning following multisensory training (Zilber et al. 2014; La Rocca et al. 2020). One discrepancy between previous results could be explained by differences in the method of analysis as most previous studies reporting strong frontal effects relied on multivariate analyses. For instance, one recent study (Mihalik and Noppeney 2020) found the implication of the lateral prefrontal cortex in encoding causal decision in a causal inference task but only when relying on a multivariate decoding approach. Alternatively, as seen in the Multisensory Correlation Detector, cause and order result in two distinct signals, which may follow separate paths: as a result, while the output serving causal judgment (MCD_{CORR}) is readily captured by the model, the output serving temporal order (MCD_{LAG}) may require additional steps to yield behavioral decision. This observation is consistent with the fact that temporal order and simultaneity judgments do not readily correlate intra-individually (Recio et al. 2019). It is also consistent with previous MEG work which, using both univariate and multivariate analyses, showed the implication of prefrontal cortices in the temporal aspects of multisensory integration (Zilber et al. 2014; La Rocca et al.

2020). Further, the implication of prefrontal cortices in temporal order judgements is also consistent with their known role in the temporal sequencing of events (Fuster 2001) and in decision-making tasks that involve a mixture of feedforward and feedback processes (Siegel et al. 2015).“